# Gap-filling a spatially-explicit plant trait database: comparing imputation methods and different levels of environmental information.

Rafael Poyatos[1,2], Oliver Sus[3], Llorenç Badiella[4], Maurizio Mencuccini[1,5], Jordi Martínez-Vilalta[1,6]

[1] CREAF, E08193 Bellaterra (Cerdanyola del Vallès), Catalonia, Spain
[2] Laboratory of Plant Ecology, Faculty of Bioscience Engineering, Ghent University, Coupure links 653, 9000 Gent, Belgium
[3] EUMETSAT, Eumetsat Allee 1, 64295 Darmstad, Germany
[4] Servei d'Estadística Aplicada, Universitat Autònoma de Barcelona, Cerdanyola del Vallès 08193, Barcelona, Spain
[5] ICREA, Barcelona, Spain
[6] Universitat Autònoma de Barcelona, E08193 Bellaterra (Cerdanyola del Vallès), Catalonia, Spain

*Correspondence to*: Rafael Poyatos (r.poyatos@creaf.uab.es)

**Abstract.** The ubiquity of missing data in plant trait databases may hinder trait-based analyses of ecological patterns and processes. Spatially-explicit datasets with information on intraspecific trait variability are rare but offer great promise in improving our understanding of functional biogeography. At the same time, they offer specific challenges in terms of data imputation. Here we compare statistical imputation approaches, using varying levels of environmental information, for five plant traits (leaf biomass to sapwood area ratio, leaf nitrogen content, maximum tree height, leaf mass per area and wood density) in a spatially-explicit plant trait dataset of temperate and Mediterranean tree species (IEFC dataset for Catalonia, north-east Iberian Peninsula, 31900 km$^2$). We simulated gaps at different missingness levels (10% – 80%) in a complete trait matrix, and we used overall trait means, species means, k-nearest neighbours (kNN), ordinary and regression kriging and multivariate imputation using chained equations (MICE) to impute missing trait values. We assessed these methods in terms of their accuracy and of their ability to preserve trait distributions, multi-trait correlation structure and bivariate trait relationships. The relatively good performance of mean and species mean imputations in terms of accuracy masked a poor representation of trait distributions and multivariate trait structure. Species identity improved MICE imputations for all traits, whereas forest structure and topography improved imputations for some traits. No method performed best consistently for the five studied traits, but, considering all traits and performance metrics, MICE informed by relevant ecological variables gave the best results. However, at higher missingness (> 30%) species mean imputations and regression kriging tended to outperform MICE for some traits. MICE informed by relevant ecological variables allowed to fill the gaps in the IEFC incomplete dataset (5495 plots) and quantify imputation uncertainty. Resulting spatial patterns of the studied traits in Catalan forests were broadly similar when using species means, regression kriging or the best-performing MICE application, but some important discrepancies were observed at the local level. Our results highlight the need to assess imputation quality

beyond just imputation accuracy, and show that including environmental information in statistical imputation approaches yields more plausible imputations in spatially-explicit plant trait datasets.

## 1 Introduction

Trait-based ecology has emerged in recent years as one of the most active ecological sub-disciplines, specially in plant ecology (Westoby & Wright, 2006; Violle et al. 2007). The move from a taxonomic perspective of biodiversity towards a focus on continuous axes of functional variation holds promise for greater generalisation, synthesis and predictive ability in ecology (Funk et al. 2016; Shipley et al. 2016). As a result, plant ecologists have increasingly embraced trait-based approaches because they may be specially suited to study plant strategies (Reich, 2014), community assembly and dynamics (McGill et al. 2006) or ecosystem functioning, particularly in the context of global environmental change (Reichstein et al. 2014). But trait-based ecology is also unquestionably thriving because of the increasing availability and reliability of plant trait data (Kattge et al. 2011).

Plant trait databases compiled from multiple individual contributions lack a common design and inevitably result in sparse data matrices (e.g. Jetz et al. 2016). Complete-case analyses (i.e., data analyses using only sampling units with complete data availability) entail a reduced sampling size, which complicates community-level studies (Pakeman, 2014) and limit the spatial coverage of trait maps usable in trait-based models of ecosystem function. Data deletion may also bias parameter estimates (e.g., in trait relationships) if the data are not missing completeley at random (MCAR; Little & Rubin, 2002; Nakagawa & Freckleton, 2008). Imputation (i.e., gap-filling) of missing data with plausible values has the potential to overcome some of these limitations, but has has only relatively recently started to be widely advocated in ecology (Nakagawa & Freckleton, 2008). It should be noted, however, that imputation may not be recommended in certain studies (Blonder 2016).

Single imputation methods replace a missing datum by one value and proceed with the analysis as if the imputed data had been observed (Nakagawa & Freckleton, 2008). Within these approaches, species mean or median imputation are probably the most widely used methods in ecology, but they ignore the variance of the imputed variables. Model-based imputation methods use other variables in the dataset to impute missing data, but they substantially alter the univariate trait distributions and the covariance structure of the dataset (Gelman & Hill, 2007). Approaches such as $k$-nearest neighbour (kNN) or machine-learning methods (Stekhoven & Bühlmann, 2012) may be more appropriate to impute multivariate datasets, preserving their covariance structure (Eskelson et al. 2009; Penone et al. 2014). In a multiple imputation framework, $m$ imputed datasets are obtained through simulation and may be jointly analysed to provide parameter estimates that take into account the uncertainty introduced by the imputations themselves (e.g. Fisher et al. 2003). Some multiple imputation

techniques, such as multivariate imputation using chained equations (MICE) may be specially well-suited to preserve the original structure and distribution of multivariate datasets (van Buuren & Groothuis-Oudshoorn, 2011; van Buuren, 2012).

While forest inventories have adopted statistical imputation methods for some time, as for example the kNN methods (Eskelson et al. 2009 and references therein), imputation methods have only recently started to be used in trait-based ecology

(Baraloto et al. 2010; Pyšek et al. 2015). Complex imputation methods such as kNN, MICE or random forests generally outperform overall mean or species mean imputations (Penone et al. 2014; Taugourdeau et al. 2014). In earlier applications of these methods, it has been common to assume that interspecific trait variability was dominant, compared to intraspecific variability. The strong phylogenetic signal may then be sufficient to impute species-averaged trait values using taxonomic information (Swenson, 2014). However, intraspecific variability in plant traits may be substantial (Siefert et al. 2015; Vilà-

Cabrera et al. 2015) and imputation methods that use environmental information may be more appropriate when assessing trait relationships and trait-environment covariance in a spatially explicit context. Biotic or abiotic variables other than the trait matrix of interest can be included in imputation algorithms as auxiliary variables to reduce imputation bias (Azur et al. 2011; Rezvan et al. 2015). Geostatistical methods of spatial interpolation can also be used with (e.g. regression kriging) or without (e.g. ordinary kriging) auxiliary variables (e.g. Hengl et al. 2007).

Additional challenges occur in the imputation of traits in large databases. The expected declining performance of imputation methods with increasing missingness levels, may be trait- and dataset-dependent (Penone et al. 2014; Taugourdeau et al. 2014). Moreover, the impact of imputations on altering bivariate trait relationships has only been assessed for single relationships (Penone et al. 2014; Schrodt et al. 2015) and not for the multiple relevant relationships within a plant trait

dataset. Likewise, there are few studies quantifying how different imputation methods alter the multivariate covariance structure of plant trait datasets (Schrodt et al. 2015).

Our overarching aim here is to assess the performance of different imputation methods to fill simulated gaps at different missingness levels in a spatially-explicit plant trait dataset (IEFC, Ecological and Forest Inventory of Catalonia, north-east

Iberian Peninsula). We imputed these missing data using single imputation (kNN), multiple imputation (MICE) and geostatistical approaches (ordinary and regression kriging, OrdKrig and RegKrig, respectively), and compared the imputations with baseline scenarios of overall mean and species mean imputation. Imputation performance was assessed in terms of accuracy, univariate trait distributions, multivariate trait structure and deviations in trait relationships. Our specific objectives are: (i) to test which imputation method (overall mean imputation, kNN, MICE, OrdKrig) performed best when

relying only on plant trait data; (ii) to assess the impact of including additional predictors (i.e. environmental information such as species identity, climate, forest structure, topography, lithology and sampling date) in MICE and kNN imputations; (iii) to compare the performance of kNN, MICE and RegKrig using optimum levels of environmental information (i.e. the best set of predictors in objective ii); and, finally, (iv) to apply the best performing method to fill the gaps in a major subset

of the IEFC database to obtain 'continuous' maps of plant traits for the main forest species across a relatively large Mediterranean region.

## 2. Methods

### 2. 1 Study area

The study area is the entire territory of Catalonia (31900 km$^2$), in the north-east Iberian peninsula. Catalonia has 38% forest cover (1.2 x 10$^6$ ha) and forests are largely dominated by species belonging to the Pinaceae and Fagaceae families. We selected 13 tree species, including 6 *Pinus* spp., 5 deciduous and evergreen *Quercus* spp., *Abies alba* and *Fagus sylvatica*, which altogether cover >90% of the forested area in Catalonia (see Supplement S1, Fig. S1).

### 2. 2 Data

Plant trait and forest data were retrieved from the Ecological and Forest Inventory of Catalonia (IEFC), carried out between 1988 and 1998 (Gracia et al. 2000−2004). A complete description of the sampling scheme and methods used to measure plant traits in the IEFC can be found in the Supplement S1. The subset of the IEFC limited to the 13 study species, hereby called 'IEFC incomplete dataset', included 5495 plots. Forest structure, lithology and sampling information for each plot were retrieved from the IEFC database, whereas climate data were obtained from the Climatic Digital Atlas of Catalonia, with a spatial resolution of 180 m (Ninyerola et al. 2000).

We selected five plant traits (leaf mass per area, LMA, mg cm$^{-2}$; leaf nitrogen per unit mass, $N_{mass}$, %mass; maximum tree height, $H_{max}$, m; wood density, WD, gm cm$^{-3}$; leaf biomass to sapwood area ratio, $B_L$:$A_S$, t m$^{-2}$) that are used to describe major plant functional strategies (Westoby et al. 2002; Wright et al. 2004; Chave et al. 2009; Laforest-Lapointe et al. 2014). In Catalan forests, four of these traits (LMA, $N_{mass}$, $H_{max}$, WD) mostly vary between families (Pinaceae and Fagaceae) and within species (Vilà-Cabrera et al. 2015). The missing data patterns in this trait data matrix shows a much higher percentage of missing data (hereafter, 'missingness) for foliar traits, corresponding to a less intense sampling of these traits (Fig. 1). These *intentional* missing data (van Buuren 2012) would correspond to a *planned missing data design*, where missingness at random (MAR) is deliberately applied (Nagakawa 2015).

### 2.3 Experimental design

All data manipulations, imputations and statistical analyses were performed with the R programming language (R Core Team, 2015). We created a subset of the IEFC incomplete dataset only including those plots (N = 630) where all 5 traits had been measured on the dominant species ('IEFC complete dataset'). In this dataset, we randomly deleted measured values at different probability levels (10%, 20%, 30%, 50% and 80%) and independently for each trait, thus the missing data

artificially introduced are missing completely at random (MCAR). This data deletion was replicated, to yield 30 simulated datasets for each missingness level (Fig. 1). Hence, the different imputation methods were assessed on 150 datasets (5

missingness levels x 30 replicates).

We ran different single and multiple imputation algorithms (see *2.4 Imputation methods*) to fill the gaps in the trait data of the simulated incomplete datasets. Single imputation methods yield $m = 1$ imputed dataset per simulated dataset and here we set the multiple imputation methods to yield $m = 5$ datasets per simulated dataset to incorporate imputation uncertainty. Prior

to the calculation of different performance metrics for each dataset, trait values in multiply imputed datasets were averaged (Penone et al. 2014). Performance metrics were assessed using the measured values of each trait in the IEFC complete dataset (see *2.6. Statistical evaluation of the imputations)*. Note that each imputed *dataset* contains both measured and gap-filled data, but the expression 'imputed *values'* refers only to gap-filled data.

### 2.4 Imputation methods

We compared imputation methods with different degrees of complexity. We used two simple approaches to provide baseline imputations; Mean imputation ('Mean') filled missing data using the overall mean value for each trait and species mean imputation ('Spmean') replaced missing values with trait means computed for each species. Because of the spatial nature of the dataset, we also tested two geostatistical approaches, ordinary kriging ('OrdKrig') and regression kriging ('RegKrig'). Lastly, we also used two methods designed to handle multivariate datasets:  *k*-nearest neighbour imputation ('kNN') and

MICE (Multivariate Imputation using Chained Equations).

Ordinary kriging calculates a weighted average of nearby observations to predict values of a target variable in an unmeasured location, with weights that minimize prediction error and depend on spatial structure of the target variable via a variogram model (Hengl et al. 2007). Regression kriging combines a deterministic model of the target variable as a function

of auxiliary variables with kriging applied to fit the residuals (Hengl et al., 2007). We included climate and forest structure variables in the model used for regression kriging imputations (cf. '*Comparative assessment of imputation methods*'), but not species identity, because there were not enough data to generate the experimental variograms for some of the less common species for all the simulations. We performed all kriging imputations with the 'autoKrige' function in the automap R package. This function tests different variogram models and applies the best-fit variogram model for kriging (Hiemstra et al.

2009).

   The kNN method calculates a multivariate distance using only non-missing variables, selects the *k*-nearest plots with measured values for the target missing trait and aggregates these *k* neighbouring values to replace the missing value (R package VIM; Templ et al. 2013). We selected $k = 7$ and median aggregation after some preliminary tests (Supplement S2, Fig. S2). We also analysed how the inclusion of auxiliary variables in the distance calculation affected imputation

performance (see *2.5 Comparative assessment of imputation methods*).

The MICE algorithm (van Buuren & Groothuis-Oudshoorn, 2011; van Buuren, 2012) sequentially and iteratively imputes incomplete data, variable by variable, using individual imputation models conditionally specified by the user. One cycle through all the imputed variables is one iteration and MICE performs $t$ iterations in $m$ parallel streams, generating $m$ multiple imputations (Supplement S3). We set $t = 20$ to ensure convergence and to minimise the effects of imputation order (van Buuren 2012). Stochasticity is introduced in the imputation process because the parameters of the univariate imputation models are drawn from their posterior distributions, obtained using a Gibbs sampler (van Buuren, 2012). Assessments of imputation methods in the ecological literature have not tested the impact of the choice of univariate imputation models within MICE (Penone et al. 2014, Taugourdeau et al. 2014). Here we showed that predictive mean matching (PMM), the default algorithm in the mice package, performed well compared to alternative methods (Supplement S3, Fig. S3, S4). Therefore, we used MICE with PMM as the univariate imputation model, also because it is robust to non-normality and preserves non-linear relationships between variables (Morris et al., 2014). Several parameters must be tuned to specify the imputation models in the R implementation of MICE (mice package) to yield reliable imputations (van Buuren & Groothuis-Oudshoorn, 2011). The specific settings used in this study are assessed in Supplement S3 (Fig. S3, S4). Please note that we will use the uppercase acronym 'MICE' to refer to the technique in general and the lowercase acronym 'mice' to refer to a particular application in this study.

## 2.5 Comparative assessment of imputation methods

We conducted three methodological comparisons of imputation performance. A first exercise compared 'Mean', 'OrdKrig', 'kNN' and 'mice' imputations. 'Mean' imputations used only the information on the target trait, 'OrdKrig' additionally used the spatial coordinates and 'mice' and 'kNN' included only the information in the trait matrix.

A second exercise assessed in detail the impact on trait imputation of including additional environmental information as auxiliary variables in MICE and kNN. We focused our detailed analysis on MICE only but we also made a simplified comparison between kNN and MICE (cf. next paragraph). The auxiliary variables we considered were species identity, a set of climatic variables (mean annual temperature, annual thermal amplitude, both in °C), a set of forest structure variables (total aboveground biomass [T ha$^{-1}$] and stem density [stems ha$^{-1}$]), a set of topographical variables (county, elevation [m.a.s.l.], slope [°] and aspect), lithology (calcareous, non-calcareous or undetermined) and sampling month. These predictors were complete and they did not need to be imputed themselves. The selection of the specific variables describing climate and forest structure was based on a recent analysis of trait variation in the same IEFC dataset (Vilà-Cabrera et al. 2015). We further added topographical variables, lithology and sampling month given that they may influence some trait values (Niinemets, 2015; Simpson et al. 2016). Species identity ('s'), climate ('c') and forest structure ('t') were introduced

in a factorial design to identify those combinations of variables leading to improved imputations. Because we expected them to play a secondary role in explaining trait variability, topography ('p'), lithology ('l') and sampling month '(m') were sequentially added to MICE and kNN imputations using species, climate and forest structure. Topography included spatial structure through the 'county' variable; preliminary tests using coordinates instead of 'county' did not show better results. Thus, 'mice_ctsplm' was the MICE application with the highest level of environmental information (Fig. 1).

The third exercise compared species mean imputations ('Spmean') with MICE and kNN using two different levels of auxiliary variables: (i) only species identity ('mice_s' and 'kNN_s') and (ii) the level of auxiliary variables which performed best overall in the second exercise. In this same exercise, we also compared the previous approaches with 'OrdKrig' and regression kriging ('RegKrig') imputations. This third exercise thus compares a baseline scenario of 'Spmean' with imputation approaches informed either by species identity only or by an optimum level of environmental information.

## 2.6 Statistical evaluation of the imputations

Imputation performance was evaluated by comparing the imputed datasets with the complete, original dataset. A first set of metrics, Normalised Root Mean Square Error (NRMSE) and Kling-Gupta Efficiency (KGE), was calculated only for those values that had been randomly deleted and subsequently gap-filled. We tested whether the distribution of imputed and original trait values differed using a two-sample Kolmogorov-Smirnov test, which tests the null hypothesis that two samples are identically distributed.

For each simulated dataset and trait, we calculated the Normalised Root Mean Square Error (NRMSE) as a measure of accuracy:

$$NRMSE = \sqrt{\frac{mean\left[\left(y_{imp} - y_{obs}\right)^2\right]}{var\left(y_{obs}\right)}} \text{(Eq. 1)}$$

where $y_{imp}$ and $y_{obs}$ represent the vectors of imputed and observed values for a given trait, respectively. Values of NRMSE approaching zero denote a better performance of the imputation method. We also calculated a dataset-averaged NRMSE by averaging the values of NRMSE for all the traits.

We further assessed imputation performance for each trait by using KGE, a goodness-of-fit measure originally developed for hydrological models, as implemented in the R package hydroGOF (Zambrano-Bigiarini, 2014):

$$KGE = 1 - \sqrt{\left(r-1\right)^2 + \left(vr-1\right)^2 + \left(\beta-1\right)^2} \text{(Eq. 2)}$$

where $r$ is the Pearson correlation coefficient between observed and imputed values, $vr$ is the ratio of the standard deviations between imputed and observed values and $\beta$ is the ratio of imputed and observed means. The KGE range is [-∞,1], with

higher values indicating better imputation performance. KGE jointly assesses correlation, bias and difference in variability between imputed and observed values, and it is therefore a powerful, synthetic indicator of imputation quality in spatially-explicit datasets.

A second set of metrics compared the whole complete trait dataset $Y_{obs}$ with the whole imputed dataset $Y_{imp}$ (i.e. including observed and gap-filled trait values). The deviations from the original multi-trait correlation structure of the trait dataset were quantified by comparing the correlation matrices of the original and imputed datasets using the following index:

$$\Delta cormat = \sum \left| L\left[cor\left(Y_{obs}\right)\right] - L\left[cor\left(Y_{imp}\right)\right] \right| \text{(Eq. 3)}$$

Where $L\left[cor\left(Y_{obs}\right)\right]$ denotes the lower triangular part of the correlation matrix of the observed dataset and $L\left[cor\left(Y_{imp}\right)\right]$ denotes the lower triangular part of the correlation matrix of the imputed dataset. $\Delta cormat$ is indicative of the aggregated absolute difference between correlation matrices. Note that some traits were log-transformed before the calculation of the corresponding correlation matrix, following Vilà-Cabrera et al. (2015).

We also tested the impact of the imputation algorithms on selected bivariate trait relationships: $H_{max}$−WD and $N_{mass}$−LMA (log-transformed when necessary); as the correlation coefficients ($r$) of these relationships were >0.3 in absolute value and were highly significant in the complete dataset. We quantified the relative difference between the complete and the imputed datasets by calculating:

$$\% \Delta r = 100 \cdot \left|r_{obs} - r_{imp}\right| / \left|r_{obs}\right| \text{(Eq. 4)}$$

Throughout the paper, we show violin plots representing the median and the distribution of each performance metric as a function of missingness levels, but we only graphically display the 10%, 30%, 50% and 80% levels, for ease of visualisation. We modelled imputation metrics in a linear mixed-effects model (LME) as a function of the interaction between imputation method and missingness, with dataset replicate as random effect. The LME model was fitted using the nlme package in R (Pinheiro et al., 2012) and pairwise comparisons of model coefficients were performed using the lsmeans and lstrends functions in the lsmeans package (Lenth, 2016).

## 2.7 Imputing traits for the main forest species in Catalonia

Finally, we applied three imputation methods to gap-fill and map the five traits across all the plots in the 'IEFC incomplete dataset'. We chose 'Spmean', as the most widely used imputation method in trait-based studies 'RegKrig, as a reference geostatistical approach including auxiliary variables and 'mice_ctsp', as the best method overall, considering all traits and performance metrics (see *3. Results and discussion*). We ran 'mice_ctsp' setting *m = 50* (i.e. 50 imputations per missing value), a value closer to the missingness rate, as recommended for final MICE applications (van Buuren, 2012).

## 3. Results and discussion

### 3.1 Mean imputations compared to MICE and kNN imputations using only trait information

In general, 'mice' and 'kNN' imputations resulted in more accurate imputations in terms of NRMSE than 'Mean' at low missingness rates (10%). However, at moderate and high missingness both 'mice' and 'kNN' were comparable to or outperformed by 'Mean', and specially by 'OrdKrig' (Fig. 2, Fig. S5). 'OrdKrig' was the best-performing method, in terms of NRMSE, at missingness $\geq 50\%$ ($P < 0.05$), although for three traits its performance was indistinguishable from that of 'Mean' imputations ($N_{mass}$, $H_{max}$, LMA; $P > 0.05$). Even if 'Mean' imputations imply the rather naive assumption that species identity may be unknown in a given dataset, it is nonetheless useful to compare 'Mean' imputations against 'mice' and 'kNN', which use the full trait matrix for prediction. In this case, trait covariation did not improve imputations at high missingness; recent assessments also report that the performance of MICE and kNN notably declines when missingness is $\geq$ 30% (Penone et al. 2014; Taugourdeau et al. 2014). Therefore, our results for 'OrdKrig', compared to those for 'mice' and 'kNN', show that spatial structure, rather than trait covariation, may provide more accurate trait imputations when gaps are frequent (Fig. 2, Fig. S5, S6).

As expected (Gelman and Hill, 2007), 'Mean' imputation severely altered trait distributions (Fig. S6), and introduced larger errors in selected trait correlations (Fig. 3). 'Mean' imputations also tended to cause larger deviations in the correlation matrix (Fig. S5). 'kNN' showed the lowest $\Delta cormat$ below 50% missingness ($P > 0.05$) but its performance declined at high missingness (Fig. S5). In contrast, 'mice' closely tracked observed trait distributions (Fig. S6), introduced the least error in trait correlations under high missingness levels (Fig. 3; $P < 0.05$) and yielded low $\Delta cormat$ at extreme missingness levels (Fig. S5). Recent results also show that kNN tends to introduce larger bias in bivariate trait relationships compared to MICE (Penone et al. 2014). 'OrdKrig' imputations altered distributions and trait correlations more than 'mice' (Fig. 3, Fig. S6), but they performed similarly in terms of $\Delta cormat$ at all missingness levels (Fig. S5).

## 3. 2 MICE imputations using different levels of environmental information

Introducing auxiliary variables as predictors improved MICE performance substantially but these improvements were dependent on the specific predictor set and trait (Fig. 4). Species identity increased KGE for all traits (Fig. 4) and it was the major predictor for $N_{mass}$, LMA and WD, as all MICE applications with species identity performed significantly better than those not including it (Fig. 4; $P<0.05$). Forest structure notably improved imputations for $H_{max}$ and for $B_L:A_S$, particularly at missingness $\geq 50\%$ ($P <0.05$). Climate only produced significant increases in KGE (i.e., compare 'mice' with 'mice_c' in Fig. 4) for $H_{max}$ and WD ($P<0.05$). Our results are in line with the distinct role of phylogeny and environmental variables as drivers of trait variability recently observed for the same tree species using the IEFC (Laforest-Lapointe et al. 2014; Vilà-Cabrera et al. 2015). One of these studies shows that, after controlling for family (Pinaceae and Fagaceae), environmental variables only explained a substantial fraction of the variability for $H_{max}$, they explained very little variability for LMA and WD and played no role in explaining $N_{mass}$ (Vilà-Cabrera et al. 2015).

Including topography in MICE imputations only substantially improved $B_L:A_S$ imputations (compare 'mice_cts' with 'mice_ctsp', $P<0.05$), probably because the leaf area used in $B_L:A_S$ calculations are obtained from county-level allometries, and county is one of the variables included in the topography predictor set (see *2. Methods* and Supplement S1). Nevertheless, introducing sampling month in the predictor sets did not appreciably improve MICE imputations in terms of KGE (Fig. 4), despite that phenological variation has been reported for some foliar traits (Niinemets, 2015; but see Fajardo & Siefert, 2016). Lithology did not appreciably improve MICE imputations, in contrast with the reported influence of soil pH on some foliar traits (Maire et al. 2015; Simpson et al. 2016).

At high missingness ( $\geq 50\%$), 'mice_ctsp' (including climate, forest structure, species and topography) was always within the best-performing methods ($P<0.05$), except for LMA and WD at 80% missingness, according to KGE results (Fig. 4). In terms of dataset-averaged NRMSE, $\Delta cormat$ (data not shown) and preservation of trait distributions (Fig. S7), the inclusion of topography only produced a significant improvement for dataset NRMSE at 50% missingness ($P<0.05$).

Including auxiliary variables as predictors also decreased %$\Delta$r for selected trait relationships (Fig. 5). The best-performing MICE applications (lower %$\Delta$r) always included species identity and forest structure; including other auxiliary variables did not lower %$\Delta$r significantly ($P>0.05$).

Our results collectively suggest that, apart from species identity, different types of environmental information, particularly forest structure and topography, may improve statistical imputation schemes. In contrast, the role of climate, lithology and sampling month in improving imputations was comparatively minor. However, we selected 'mice_ctsp' as the method that performed best for all traits, because adding climate did not deteriorate imputation performance and not including

'topography' would worsen $B_L:A_S$ imputations. A negligible influence of climate and soil data on trait imputation in the TRY database was also recently reported (Schrodt et al. 2015). It is unclear, however, to what extent these results simply reflect the relatively poor quality of the climate and soil data generally available at regional scales.

### 3.3 Comparing imputation methods using optimum levels of environmental information

Adding auxiliary variables to calculate the distance matrix also improved kNN imputations. Values of KGE for 'kNN_ctsp'
were much higher than those observed for 'kNN' imputations (data not shown), which only included the trait data in the distance matrix. This improvement was largely driven by the inclusion of species identity; only for $B_L:A_S$ and $H_{max}$ did 'kNN_ctsp' perform significantly better than 'kNN_s' (Fig. 6; $P<0.05$). Likewise, adding climate and forest structure as auxiliary variables improved 'RegKrig' performance compared to 'OrdKrig' (Fig. 6, $P<0.05$), except for $N_{mass}$. For both, kNN and kriging methods, WD and $H_{max}$ were the traits for which these improvements were largest.


In terms of KGE, 'mice_ctsp' was the best performing method at 50% missingness for all traits, together with 'Spmean' for $N_{mass}$, and LMA and with 'RegKrig' for $H_{max}$ ($P>0.05$ for comparisons between 'mice_ctsp', 'Spmean' and 'RegKrig for these traits). However, at 80% missingness, 'mice_ctsp' only ranked first for $B_L:A_S$ whereas 'Spmean' showed the highest KGE for $N_{mass}$, LMA and WD and 'RegKrig' performed best for $H_{max}$ (Fig. 6). These results are consistent with the prominent role of
taxonomic identity in explaining variability in foliar traits and WD and with the higher predictive ability of environmental and spatial information in explaining $H_{max}$ (Vilà-Cabrera et al. 2015). The LME model showed that the rate of increase in KGE with increasing missingness was lowest for 'Spmean' in four out of five traits (Table 1). Compared to 'Spmean' and 'RegKrig', performance of MICE and kNN declined more with increasing missingness (Table 1, Fig. S8, Table S1), but MICE generally outperformed kNN (Fig. 6, 7), as already observed in a recent imputation assessment of species-level, life-
history traits (Penone et al. 2014). In terms of dataset-averaged NRMSE, 'mice_ctsp' and 'Spmean' were the best-performing methods at 50% and 80% missingness, respectively ($P<0.05$).

Kernel density plots (Fig. S9) and Kolmogorov-Smirnov tests (Fig. S10) showed that MICE produced imputations (especially 'mice_ctsp') most consistent with observed distributions at all missingness levels (Fig. S9, S10). 'Spmean' and
'OrdKrig' imputations modified trait distributions substantially, while 'kNN_ctsp' and 'RegKrig' showed an intermediate performance, but generally far from that of 'mice_ctsp' (Table 1, Fig. S8, Table S1). 'Spmean' and kriging imputations also yielded larger *Δcormat* values compared to the rest of the methods ($P<0.05$), reflecting their lower ability to maintain trait correlation structures.

For the selected trait relationships, 'mice_ctsp' showed the lowest values of %Δr at $\geq 50\%$ missingness ($P<0.05$), although for the $N_{mass}$−LMA relationship 'kNN_s' and 'mice_s' performed equally well at 50% and 80% missingness, respectively (Fig. 7; $P>0.05$). 'Spmean' imputations showed variable results, severely altering $H_{max}$−WD relationship ($P<0.05$, Fig. 7) but

showing comparable performance to 'mice_ctsp' for the $N_{mass}$−LMA relationship at 80% missingness ($P>0.05$, Fig. 7). Kriging imputations did not succeed in minimising changes in reproducing trait correlations (Fig. 7).


Using imputed or incomplete datasets did not lead to large differences in the studied trait relationships when missingness was <50% (Fig. S11, S12). However, at high missingness, using imputed datasets led to comparatively larger departures from the relationships obtained with the complete dataset, especially for the $N_{mass}$−$LMA$ relationship. No imputation method appeared to perform consistently better than others in preserving trait relationships at high missingness levels (Fig. S11, S12) and, under these conditions, using incomplete datasets appeared to correctly reproduce the observed trait relationships in the complete dataset.

### 3.4 Imputing traits for the main forest species in Catalonia

The application of 'mice_ctsp' allowed to fill the gaps in the IEFC incomplete dataset and quantify the variation among the multiple imputations, providing an estimation of the level of confidence in the imputed values for specific traits (Fig. 8). 'Spmean' and 'RegKrig' show a broadly similar spatial pattern of trait variation compared to 'mice_ctsp', although some important discrepancies between 'Spmean' and 'mice_ctsp' can be observed in the north-eastern pre-litoral and coastal area for LMA (Fig. S13). Here, 'Spmean' imputations tend to predict lower values compared to 'mice_ctsp'. These areas are mostly dominated by *Quercus suber* forests (Fig. S1), and LMA was only measured in 5 out of the 149 plots of this species present in the IEFC incomplete dataset. Therefore, as there is little information on trait covariation for the imputation of LMA in *Q. suber* plots, MICE imputations are largely based on the auxiliary variables and they yield a distinct spatial pattern of trait variation, compared to 'Spmean'. Imputations obtained using regression kriging result in more blurred spatial patterns, relative to other imputation methods (Fig. S13).

### 4 Implications and conclusions

The problem of missing data is ubiquitous in plant trait datasets of regional to global scope. Recently, ecologists have made substantial progress in (i) the assessment of the best imputation methods in trait-based applications, (ii) how these methods perform with increasing missingness, (iii) which ecological covariates aid to improve imputations and (iv) how different imputation methods impact the results of trait-based analyses (Pakeman, 2014, Taugourdeau et al. 2014, Penone et al. 2014, Schrodt et al. 2015). Most effort thus far, however, has been directed at imputing species-level trait means and all the abovementioned questions have rarely been assessed on the same dataset. Here we deal with all the previous issues simultaneously and also deal withthe spatial component of trait variability, where the intra-specific component cannot be neglected. We did not focus on differences in imputation errors across species because this issue is, to a large extent, related to the degree of trait variability explained by biotic and abiotic predictors across different taxa, which was recently reported by Vilà-Cabrera et al. (2015).

One limitation of this study is that we simulate data missing completely at random (MCAR) whereas a missing at random (MAR) assumption would have probably been more realistic given the properties of the dataset (Nakagawa 2015). However, a recent study did not show differences in trait imputation performance between these two missing data mechanisms (Penone et al. 2014). Our study assesses different imputation methods in spatial, traits datasets with multivariate missing data. Amongst the methods assessed here, MICE and kNN are the most adequate to impute multivariate datasets, as they can be used when predictors also include missing data. Kriging methods may be more difficult to apply when predictors are also missing, but we have shown that, at high missingness levels and when environmental information is lacking, they can outperform MICE and kNN. This implies that geostatistical methods may sometimes provide more accurate imputations than those using trait covariation.

Our results show that, in terms of trait prediction error, no imputation method performs best consistently for the five studied traits. However, when all performance metrics are jointly considered (i.e. errors in trait prediction, multivariate trait distribution and trait correlations), MICE informed by relevant ecological variables outperforms approaches based on trait averaging, geostatistical models and kNN methods, albeit this superiority of MICE tends to vanish at very high missingness levels. For kNN, MICE and kriging imputations we have highlighted the key role of auxiliary variables as necessary covariates to yield reliable imputations in spatially explicit settings. This result calls for the inclusion of site-specific environmental variables associated with trait data in trait databases. The importance of covariates differed across traits, but, in addition to the expected influence of species, climate and topography in predicting trait values, we also showed a prominent role of forest structure for some traits. The ongoing development of global databases of vegetation structure (e.g. Dengler *et al.* 2014) will likely enable the incorporation of stand variables in trait imputation approaches using spatial and environmental information (Butler et al. 2017).

Given the limited number of species in our study, reflecting the relatively low richness of the studied communities, taxonomic information introduced as species identity was enough to improve imputations of all studied traits. However, in studies coping with a larger set of species, phylogeny may need to be considered in the imputation models (Schrodt et al. 2015, Swenson et al. 2017). For global trait datasets, a combination of imputation with data augmentation approaches (e.g. Nakagawa & Freckleton, 2008) has been proposed to minimise potential errors in trait-driven analyses caused by incomplete and biased species sampling (Sandel et al. 2015).

Compared to other imputation approaches, MICE is well-suited to deal with multivariate missing data (i.e. MICE produce imputations when some predictors are also missing) and provides information to quantify the uncertainty associated with the imputed data (Fig. 8). MICE uses multivariate relationships in the dataset to impute missing data, and this may raise concerns about potential circularity in trait or trait-environment correlations. Despite these concerns, it has been argued that

the full inference framework based on multiply-imputed datasets would minimise circularity (van Buuren 2012). Because our comparative assessment of imputation methods is already complex, here we have focused on the initial *imputation* stage, the first step of the full process (e.g. Nakagawa & Freckleton 2008). MICE produces multiple datasets, with imputed values drawn from distributions, and these datasets can be combined in the *analysis* and *pooling* steps. The analysis step refers to the estimation of the parameters of scientific interest (e.g. a regression coefficient) for each dataset. In MICE, parameters can

be pooled across datasets to produce unbiased estimates and standard errors, providing a natural way to take into account the additional uncertainty introduced in the analysis by the presence of missing data, and to avoid circularity effects (van Buuren 2012). However, ecological studies using multiple imputation approaches usually only apply the imputation step (Baraloto et al. 2010, Paine et al. 2011, Pyšek et al 2015, Díaz et al. 2016) and do not take advantage of the multiple imputation framework to quantify the uncertainty resulting from the presence of missing data (but see Fisher et al. 2003).

Our results have important implications given that the demand for spatially explicit datasets is increasing rapidly and that species mean imputation and casewise data deletion are still widespread practices in trait-based ecology. We show that species mean imputation may result in substantial information loss that may hinder research development on important topics in functional biogeography, such as the ecological drivers and implications of intraspecific trait variability (e.g. Siefert

et al. 2015). Gap-filled multivariate trait datasets may increase the robustness of syntheses of plant form and function (Díaz et al. 2016) and trait-driven modelling approaches (Yang et al. 2015). We also show that spatially-distributed layers of environmental information may improve trait mapping, increasing spatial resolution and/or sample size in trait-driven ecosystem process models (Christoffersen et al. 2016).

**Data availability**

The IEFC complete trait dataset is available at 10.5281/zenodo.1209812.

**Author contributions**

RP, OS, JMV and MM conceived the study and all authors contributed to design the simulations and statistiscal analyses. RP, OS and LB carried out the simulations and analyses, assisted by the rest of the authors. RP wrote the paper with the contribution of all the coauthors.

**Competing interests**

The authors declare that they have no conflict of interest

**Acknowledgements**

We thank J. Vayreda, A. Vilà-Cabrera and S. Saura-Mas for their help with the IEFC dataset. This study was funded by the Spanish Ministry of Economy and Competitiveness (MINECO) through the grants CGL2013-46808-R, CGL2014-55883-JIN and MTM2015-69493-R.

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

**Table 1.** Tukey pairwise comparisons of the LME model coefficients (and 95% lower/upper confidence limits) relating trait-specific KGE (higher values of KGE imply higher performance) to increasing missingness levels for different imputation methods: species mean (Spmean), mice and kNN with species as predictor (mice_s and kNN_s, respectively), mice and kNN with species, climate, forest structure and spatial variables as predictors (mice_ctsp and kNN_ctsp, respectively), ordinary kriging (OrdKrig) and regression kriging (RegKrig). Traits: leaf biomass to sapwood area ratio, $B_L{:}A_S$ (t m$^{-2}$); leaf nitrogen per unit mass, $N_{mass}$ (%mass);  maximum tree height, $H_{max}$ (m); leaf mass per area LMA (mg cm$^{-2}$); wood density, WD, (gm cm$^{-3}$). Different letters, in alphabetical order following the increasing order of the model coefficient, denote significant differences ($P < 0.05$) in the results of multiple comparisons.

| | Coefficient | SE | df | Lower CL | Upper CL |
|---|---|---|---|---|---|
| $B_L{:}A_S$ | | | | | |
| OrdKrig | 5.27E-04 | 3.93E-04 | 148 | 2.50E-04 | 1.30E-03 a |
| RegKrig | 9.25E-04 | 3.93E-04 | 148 | 1.49E-04 | 1.70E-03 ab |
| mice_s | 1.16E-03 | 3.93E-04 | 148 | 3.79E-04 | 1.93E-03 ab |
| Spmean | 1.29E-03 | 3.93E-04 | 148 | 5.17E-04 | 2.07E-03 ab |
| mice_ctsp | 1.70E-03 | 3.93E-04 | 148 | 9.28E-04 | 2.48E-03 bc |
| kNN_s | 2.29E-03 | 3.93E-04 | 148 | 1.52E-03 | 3.07E-03 c |
| kNN_ctsp | 3.27E-03 | 3.93E-04 | 148 | 2.50E-03 | 4.05E-03 d |
| $N_{mass}$ | | | | | |
| OrdKrig | 1.14E-03 | 2.80E-04 | 148 | 5.88E-04 | 1.69E-03 a |
| Spmean | 1.41E-03 | 2.80E-04 | 148 | 8.54E-04 | 1.96E-03 a |
| RegKrig | 1.49E-03 | 2.80E-04 | 148 | 9.40E-04 | 2.04E-03 a |
| mice_s | 1.50E-03 | 2.80E-04 | 148 | 9.49E-04 | 2.05E-03 a |
| mice_ctsp | 2.33E-03 | 2.80E-04 | 148 | 1.77E-03 | 2.88E-03 b |
| kNN_ctsp | 2.38E-03 | 2.80E-04 | 148 | 1.82E-03 | 2.93E-03 b |
| kNN_s | 2.70E-03 | 2.80E-04 | 148 | 2.15E-03 | 3.26E-03 b |
| $H_{max}$ | | | | | |
| RegKrig | 7.11E-04 | 2.32E-04 | 148 | 2.53E-04 | 1.17E-03 a |
| Spmean | 7.23E-04 | 2.32E-04 | 148 | 2.65E-04 | 1.18E-03 a |
| mice_s | 1.14E-03 | 2.32E-04 | 148 | 6.84E-04 | 1.60E-03 ab |
| OrdKrig | 1.22E-03 | 2.32E-04 | 148 | 7.62E-04 | 1.68E-03 abc |
| mice_ctsp | 1.64E-03 | 2.32E-04 | 148 | 1.19E-03 | 2.10E-03 bc |
| kNN_s | 1.75E-03 | 2.32E-04 | 148 | 1.30E-03 | 2.21E-03 bc |
| kNN_ctsp | 1.77E-03 | 2.32E-04 | 148 | 1.32E-03 | 2.23E-03 c |

**LMA**

| | | | | | | |
|---|---|---|---|---|---|---|
| OrdKrig | 7.36E-04 | 1.99E-04 | 148 | 3.43E-04 | 1.13E-03 | a |
| RegKrig | 8.36E-04 | 1.99E-04 | 148 | 4.43E-04 | 1.23E-03 | a |
| Spmean | 1.24E-03 | 1.99E-04 | 148 | 8.50E-04 | 1.64E-03 | a |
| mice_s | 1.82E-03 | 1.99E-04 | 148 | 1.43E-03 | 2.21E-03 | b |
| kNN_ctsp | 2.26E-03 | 1.99E-04 | 148 | 1.87E-03 | 2.65E-03 | b |
| mice_ctsp | 2.88E-03 | 1.99E-04 | 148 | 2.48E-03 | 3.27E-03 | c |
| kNN_s | 3.20E-03 | 1.99E-04 | 148 | 2.80E-03 | 3.59E-03 | c |

**WD**

| | | | | | | |
|---|---|---|---|---|---|---|
| OrdKrig | -7.13E-04 | 1.41E-04 | 148 | -9.92E-04 | -4.35E-04 | a |
| RegKrig | 2.65E-05 | 1.41E-04 | 148 | -2.52E-04 | 3.05E-04 | b |
| Spmean | 2.94E-04 | 1.41E-04 | 148 | 1.52E-05 | 5.72E-04 | b |
| mice_s | 7.70E-04 | 1.41E-04 | 148 | 4.91E-04 | 1.05E-03 | c |
| kNN_s | 1.08E-03 | 1.41E-04 | 148 | 8.05E-04 | 1.36E-03 | c |
| kNN_ctsp | 1.14E-03 | 1.41E-04 | 148 | 8.58E-04 | 1.41E-03 | cd |
| mice_ctsp | 1.56E-03 | 1.41E-04 | 148 | 1.28E-03 | 1.84E-03 | d |

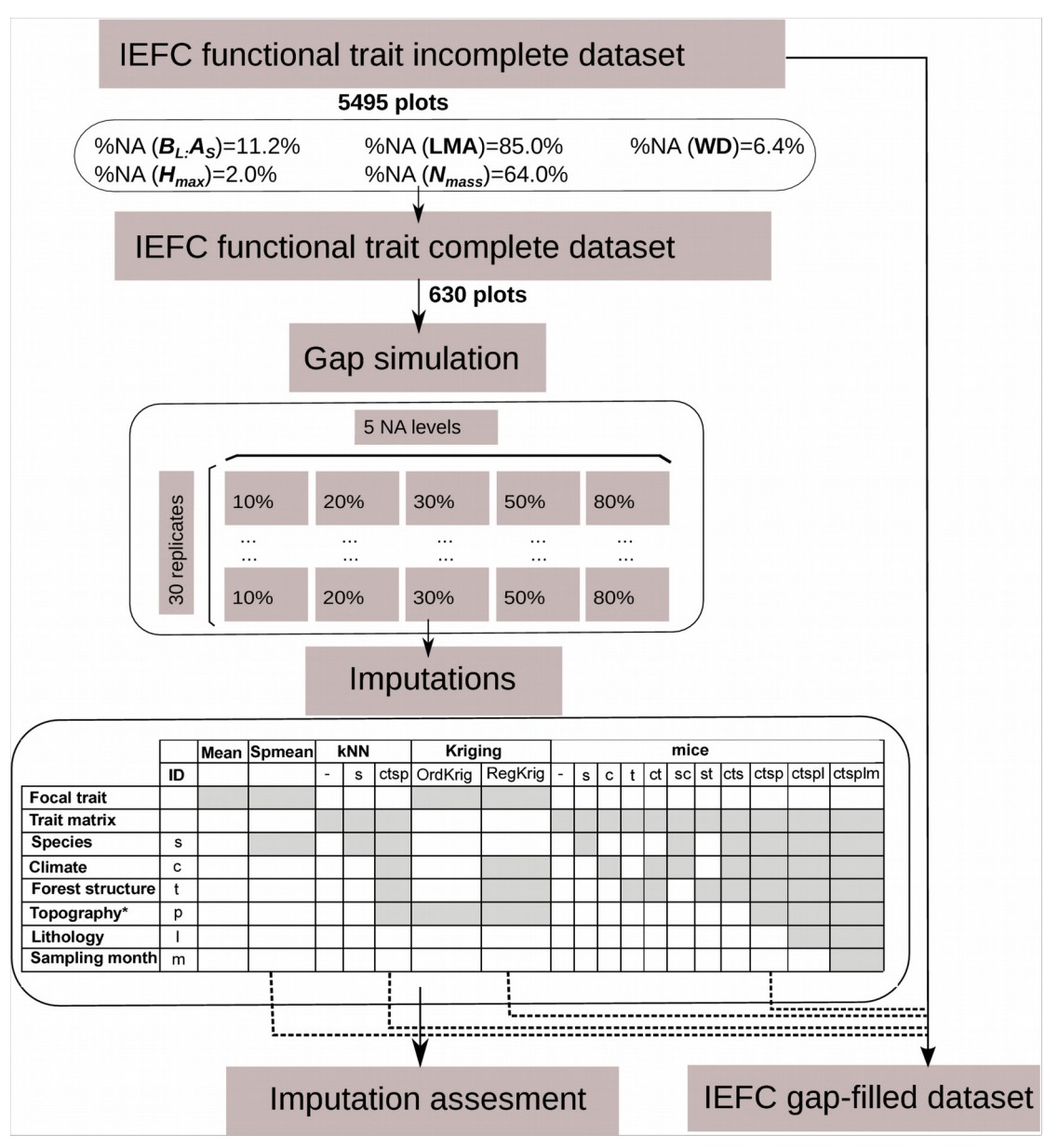

**Figure 1.** Description of the experimental design. A subset was obtained from the incomplete IEFC trait dataset containing only plots where all functional traits had been measured (complete dataset) to perform the gap simulations and the imputations. Imputation methods are described in terms of the input information used. The selected methods for the final application of imputation methods to obtain a gap-filled IEFC trait dataset are also shown.

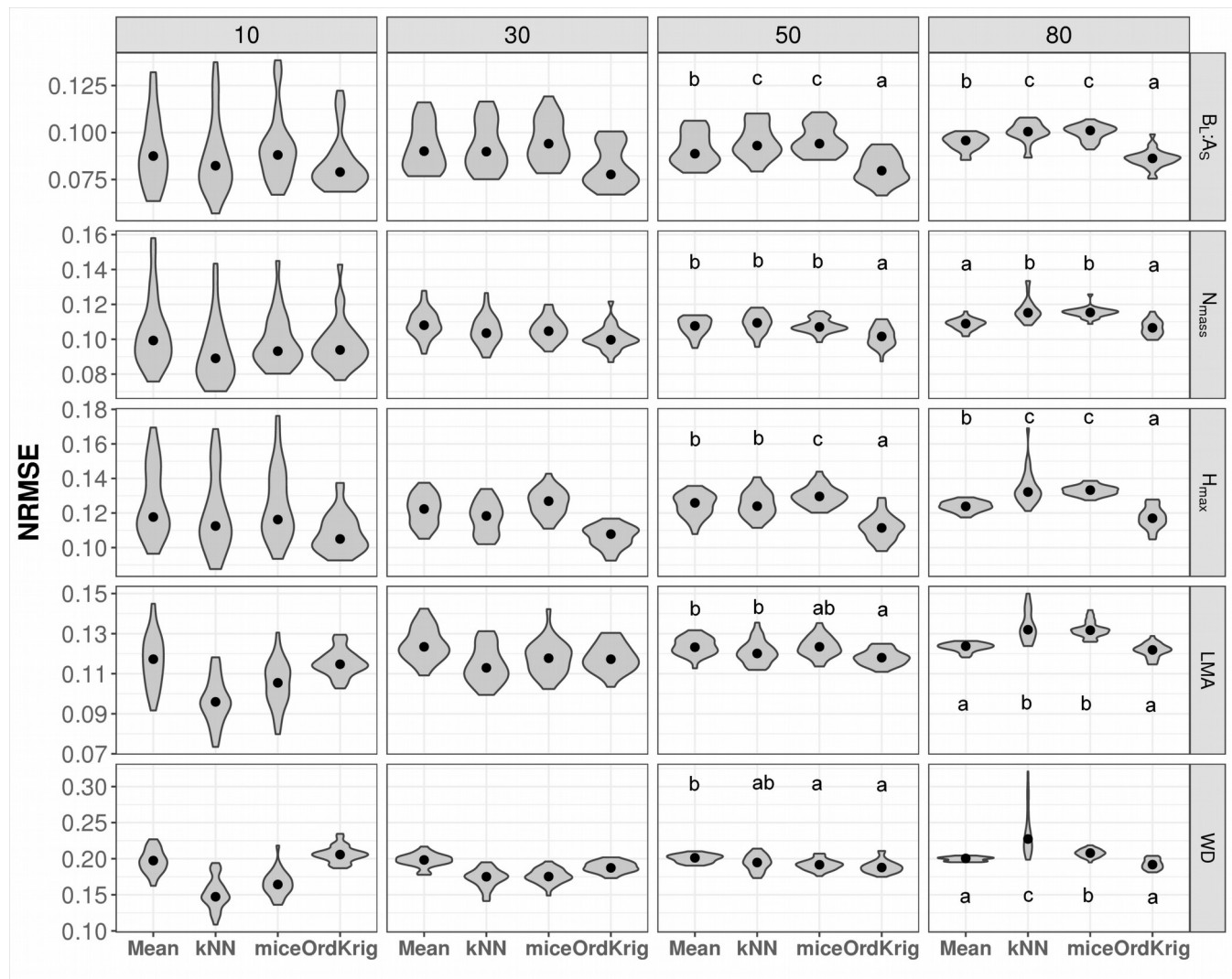

**Figure 2.** Trait-specific NRMSE at increasing missingness levels (10% to 80%) for different imputation methods: overall trait mean (Mean), mice (using only the trait matrix in the predictor set), kNN (using only the trait matrix for the distance calculation) and ordinary kriging (OrdKrig). Traits: leaf biomass to sapwood area ratio, $B_L:A_S$ (t m$^{-2}$); leaf nitrogen per unit mass, $N_{mass}$ (%mass); maximum tree height, $H_{max}$ (m); leaf mass per area LMA (mg cm$^{-2}$); wood density, WD, (gm cm$^{-3}$). Letters denote results of multiple comparisons, in alphabetical order from highest to lowest performance.

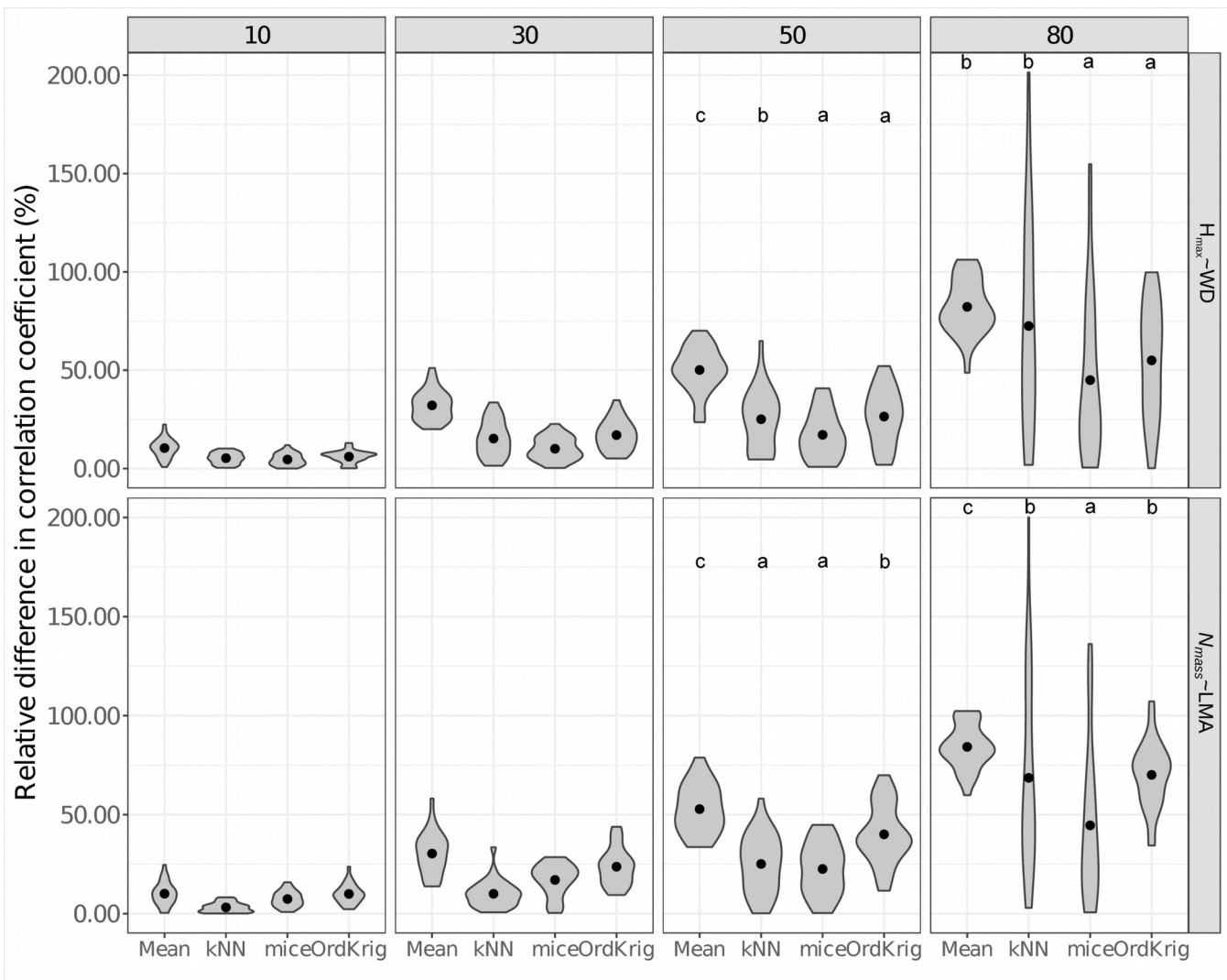

**Figure 3.** Errors in the correlation coefficient for two selected trait relationships, at increasing missingness levels (10% to 80%) and for different imputation methods: overall trait mean (Mean), mice (using only the trait matrix in the predictor set), kNN (using only the trait matrix for the distance calculation) and ordinary kriging (OrdKrig). Letters denote results of multiple comparisons, in alphabetical order from highest to lowest performance. Traits involved in the relationships are: leaf nitrogen per unit mass, $N_{mass}$ (%mass); maximum tree height, $H_{max}$ (m); leaf mass per area LMA (mg cm$^{-2}$); wood density, WD, (gm cm$^{-3}$).

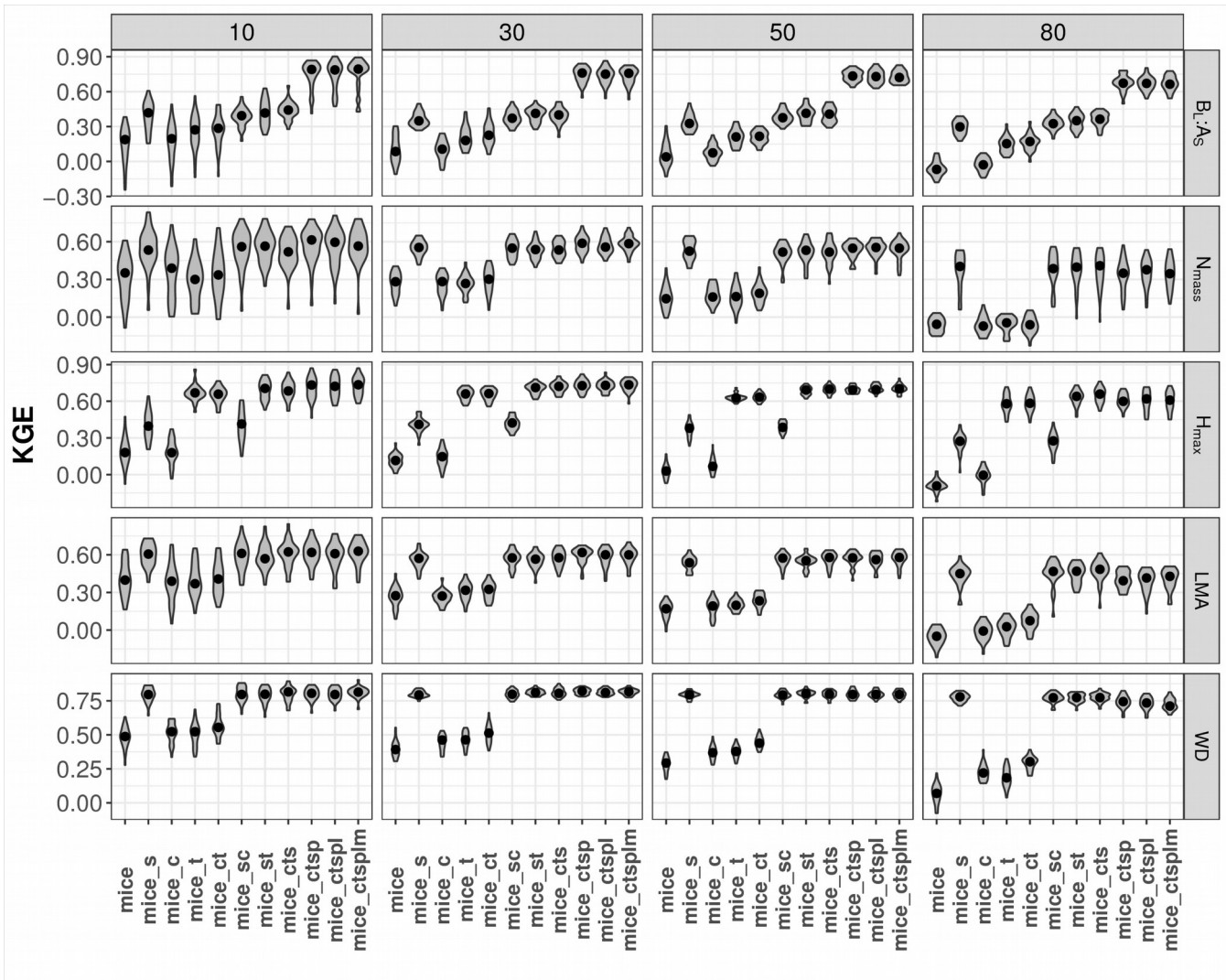

**Figure 4.** Trait-specific KGE at increasing missingness levels (10% to 80%) and for different MICE imputations using different combinations of additional predictor sets: species identity (s), climate (c), forest structure (t), sptatial structure (p), lithology (l) and sampling month (m). See Fig. 1 for an overall view of the experimental design and the Methods section for a detailed description of the variables employed in each predictor set. Traits: leaf biomass to sapwood area ratio, $B_L$:$A_S$ (t m$^{-2}$); leaf nitrogen per unit mass, $N_{mass}$ (%mass); maximum tree height, $H_{max}$ (m); leaf mass per area LMA (mg cm$^{-2}$); wood density, WD, (gm cm$^{-3}$). Higher values of KGE imply higher performance.

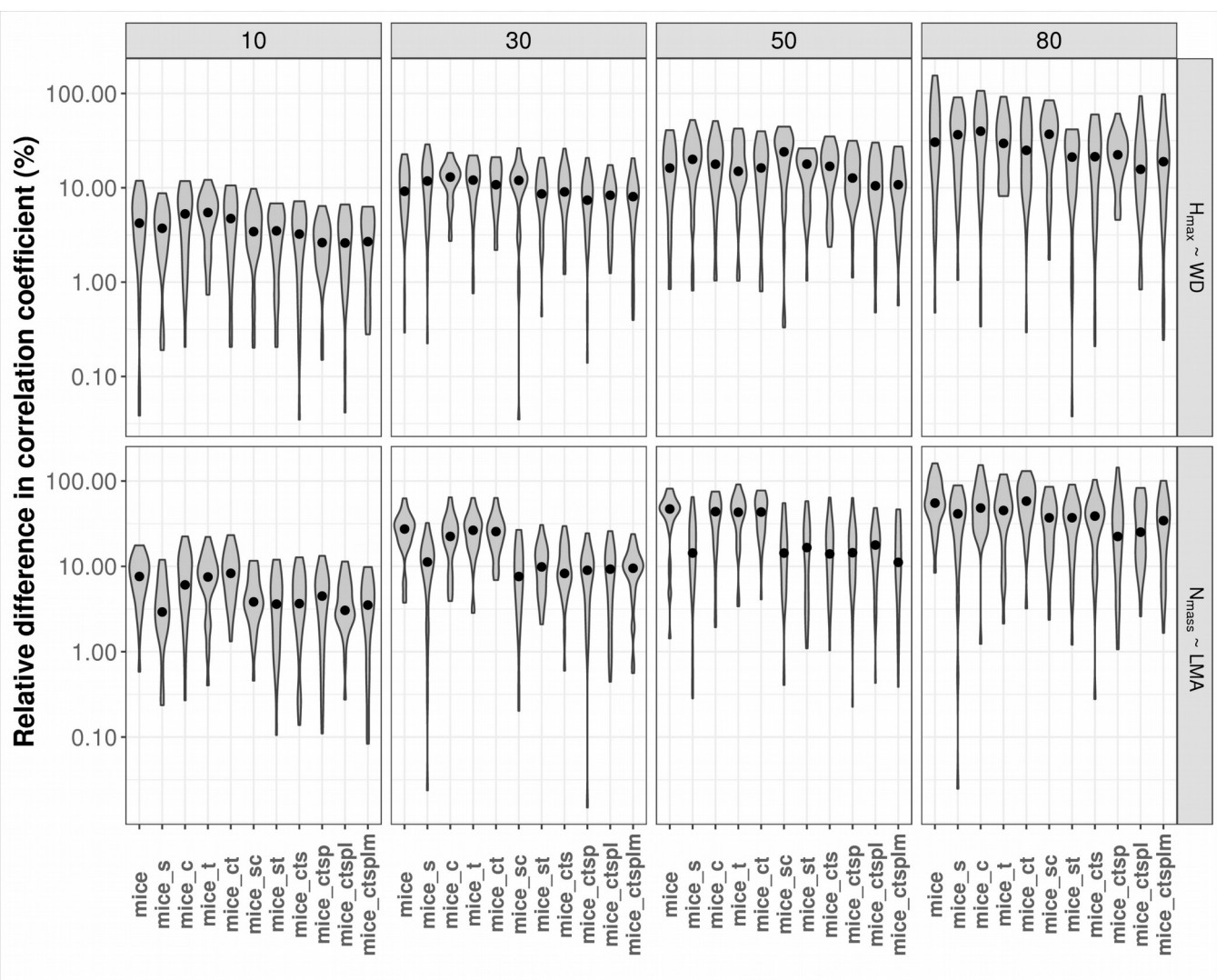

**Figure 5.** Errors in the correlation coefficient for two selected trait relationships, at increasing missingness levels (10% to 80%) and for different MICE imputations using different combinations of additional predictor sets: species identity (s), climate (c), forest structure (t), topography (p), lithology (l) and sampling month (m). See Fig. 1 for an overall view of the experimental design and the Methods section for a detailed description of the variables employed in each predictor set. Note that the y-axis is in the logarithmic scale. Traits involved in the relationships are: leaf nitrogen per unit mass, $N_{mass}$ (%mass); maximum tree height, $H_{max}$ (m); leaf mass per area LMA (mg cm$^{-2}$); wood density, WD, (gm cm$^{-3}$).

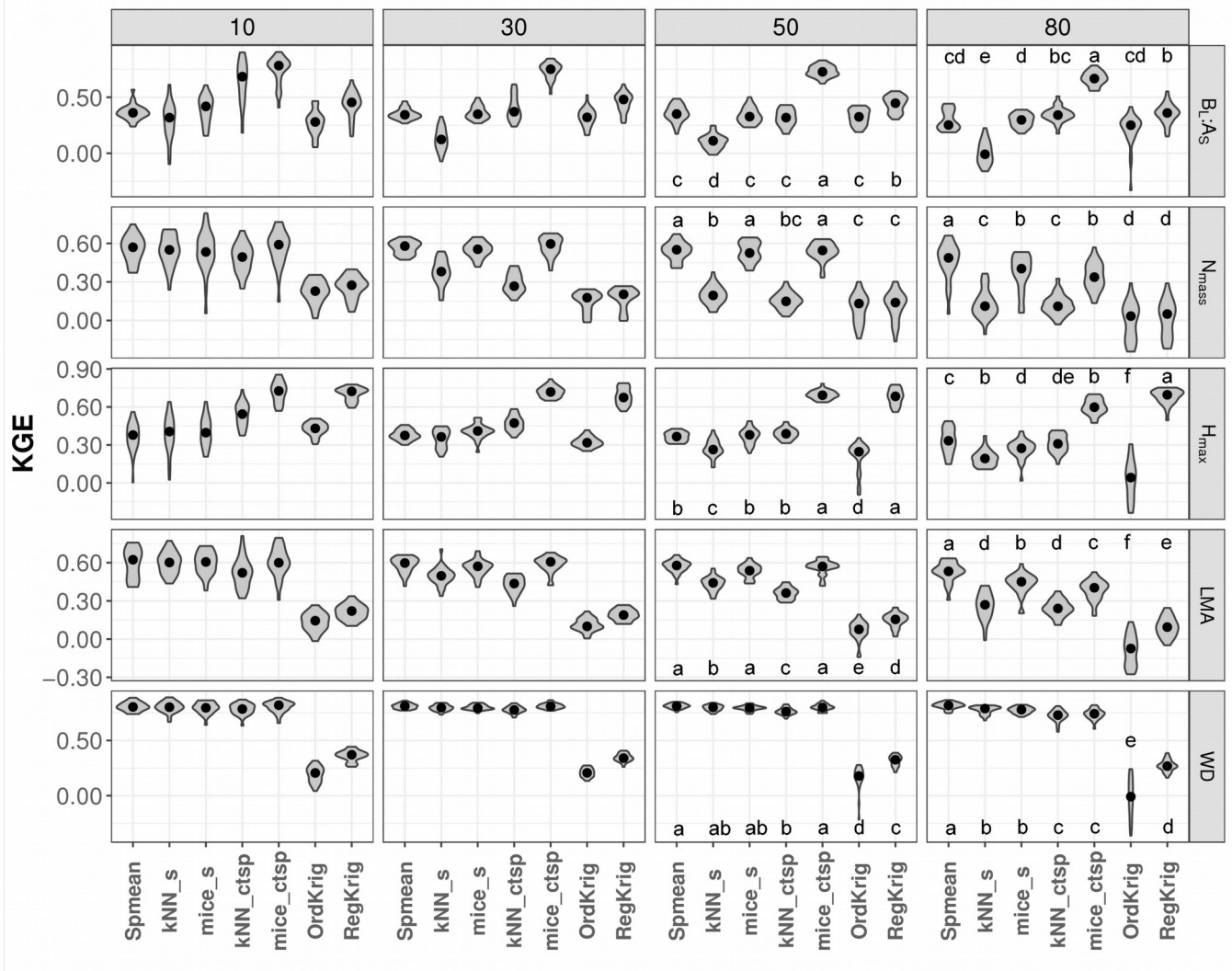

**Figure 6.** Trait-specific KGE at increasing missingness levels (10% to 80%) for different imputation methods: species mean (Spmean), mice and kNN with species as predictor (mice_s and kNN_s, respectively), mice and kNN with species, climate, forest structure and spatial variables as predictors (mice_ctsp and kNN_ctsp, respectively), ordinary kriging (OrdKrig) and regression kriging (RegKrig). Higher values of KGE imply higher performance. Traits: leaf biomass to sapwood area ratio, $B_L{:}A_S$ (t m$^{-2}$); leaf nitrogen per unit mass, $N_{mass}$ (%mass); maximum tree height, $H_{max}$ (m); leaf mass per area LMA (mg cm$^{-2}$); wood density, WD, (gm cm$^{-3}$). Higher values of KGE imply higher performance. Letters denote results of multiple comparisons, in alphabetical order from highest to lowest performance.

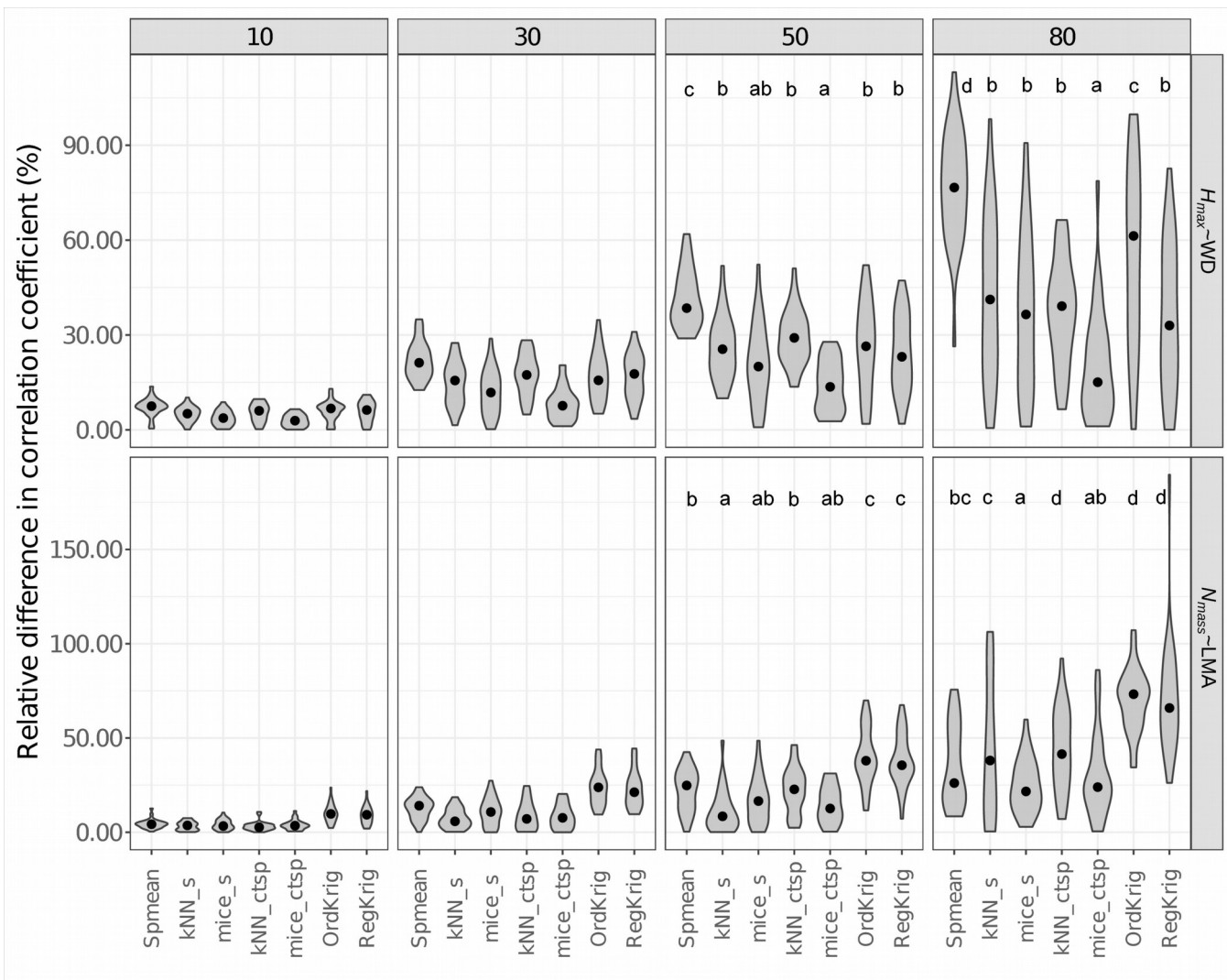

**Figure 7.** Errors in the correlation coefficient for two selected trait relationships, at increasing missingness levels (10% to 80%) and for different imputation methods: species mean (Spmean), mice and kNN with species as predictor (mice_s and kNN_s, respectively), mice and kNN with species, climate, forest structure and spatial variables as predictors (mice_ctsp and kNN_ctsp, respectively), ordinary kriging (OrdKrig) and regression kriging (RegKrig). Letters denote results of multiple comparisons, in alphabetical order from highest to lowest performance.

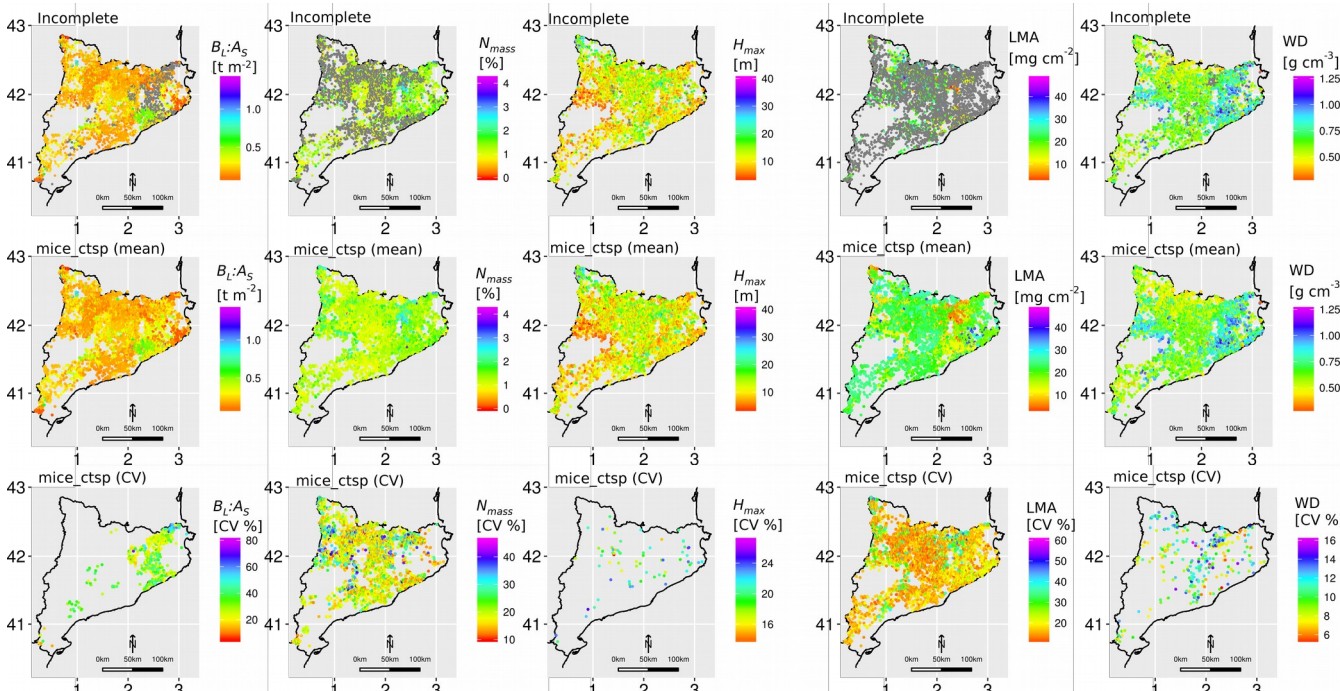

**Figure 8.** Maps with the distribution of functional traits across the selected plots in the IEFC. The first row shows the incomplete dataset, with missing values in grey. The second row shows the mean of 50 multiple imputations for each missing value using the 'mice_ctsp' approach (MICE imputation using species identity, climate, forest structure and topography as predictors). The third row shows the corresponding coefficient of variation (CV) for these multiple imputations. Note that, for the third row, only imputed values are shown and that the colour scale varies across different traits. Traits: leaf biomass to sapwood area ratio, $B_L{:}A_S$ (t m$^{-2}$); leaf nitrogen per unit mass, $N_{mass}$ (%mass); maximum tree height, $H_{max}$ (m); leaf mass per area LMA (mg cm$^{-2}$); wood density, WD, (gm cm$^{-3}$). Higher values of KGE imply higher performance.