# Peer review of "Gap-filling a spatially-explicit plant trait database: comparing imputation methods and different levels of environmental information."

_Biogeosciences, 2017_

## Referee Comment (RC1) · Anonymous Referee #1 · 10 Jan 2018

This paper compares the performance of different imputation methods for a trait dataset. Although this topic has already been studied before, the authors bring three novelties: (i) they use a spatially-explicit dataset (ii) which includes intra-specific information, and (iii) they use of a set of measures to assess imputation performance in terms of multivariate trait structure. This type of analysis is interesting, in a time where gap-filling methods are used more and more often to impute trait datasets. However, the paper would need some clarifications and better justifications, in particular for the choice of the methods. Some proper synthesis of the results is missing (shown by the total number of figures in main text and appendix) and no tests are performed, giving the general impression that none of the methods performs better than others. There is

also a number of problems in the results and figure presentations that should be fixed. Please refer to the comments below for more detail, in particular my points 7 to 10 are quite major.

Another major point is the use of "ecological information" in the imputation process. The authors show that adding this information to the imputation improves its quality, as one would expect. However, in most ecological papers, authors usually look at the relationships between traits and this "ecological information". This introduces a serious problem of circularity on any analysis using trait data imputed with "ecological information". The authors never mention this potential bias, do not suggest in which type of analyses such dataset could be used and more importantly, they do not test for this potential bias. A simple test would be to look at the relationship between traits and "ecological variables" in the complete dataset, in the datasets with missing data and in the imputed ones. This would give an idea about the importance of the bias and warn (or not, depending on the results) the users about it.

\*Abstract\*

1- L.16: "functional biogeography" is not mentioned elsewhere in the paper

2- L.16: "they offer specific challenges in terms of data imputation": these are not mentioned in the paper neither, it would actually be interesting to discuss these specific challenges

\*Introduction\*

Overall, the introduction is well written and clear, it just lacks some details on some aspects (see below).

3- Please provide a reference and explanation for the following statement L.56: "they all alter, to different degrees, the univariate trait distributions and the covariance structure of the dataset". Please also explain why multiple imputations do a better job in conserving trait distributions and covariances.

4- L60: Please cite which multiple imputation (MI) techniques are better to preserve structure and distribution of traits. MICE includes a high number of different algorithms, some accounting for variables distribution, others for interactions between variables, etc.. And actually some MICE techniques perform even worse than single imputations (e.g. see Stekhoven & Buhlmann 2012). MICE alone does not say much about the method and it would be important to specify quite early in the paper that it is MICE-PMM which is tested.

5- L 62. I would not define kNN as a sophisticated method, it's in fact quite a simple one

6- L90: sampling date is not really "ecological information", also most of the predictors mentioned were not really introduced before.

*Methods*

In general, the methods are clearly explained but some things are understandable only when reading the supplementary information. The methods are quite dense, so it is OK to have some descriptions in the appendix. However, they should be self-understandable. Figure 1 is very helpful and important. I also think that the evaluation metrics are very interesting. However, I did not understand some of the choices, which I think should be better justified (see below).

7- A quite major point is that the authors state that in the real dataset missing data is biased towards leaf traits deliberately. So data are not MCAR in the dataset (l.120). We also know that in trait datasets, values are often MAR (Nakagawa & Freckleton 2008). But the authors then remove data completely at random (L121). We also know that imputation methods are not designed for MCAR data (van Buuren & Groothuis-Oudshoorn, 2011). So why the authors chose to remove data MCAR? I would suggest to introduce data at random with the same structure as in the original dataset. This would be a fairer test than just removing data completely at random. It is important to see how the methods behave when data is MAR.

8- Another point is that there is no comparison with the dataset including missing data. The question: "should I impute or not" is an important one so it would be good to know how well the imputation methods perform (in terms of KGE and structure) when compared to just using the dataset with gaps.

9- Related to the previous, at the end, users would like to know which method is the best, considering together NRME, distributions, correlations, structure and regardless of the trait considered or the % of missing data. This could be analysed using a mixed model with the identity of the metric (NRMSE, KGE..etc) and the missing dataset identity as random factors.

10- Given that the dataset has intraspecific variation and this is presented as a novelty both in the introduction and the "implication" section, I would have expected an analysis of the error at the species level. Also, it should be noted that adding species identity assumes that interspecific variation is higher than intraspecific variation, which is OK, but should maybe be stated somewhere.

11- L. 126: why m=5? MICE authors recommend to use at least 10 iterations (van Buuren& Groothuis-Oudshoorn, 2011). I saw later that this was explained in the appendix, I think that it should be at least mentioned in the main text.

12- L.138: similar to the previous point, it would be good to state that k=7 comes from an additional analysis (described in the appendix), otherwise the "7" is quite obscure. All these omissions make the reading/methods understanding quite challenging.

13- The MICE paragraph (l.151) should give more details on what is described in appendix S4.

14- Why some variables are added sequentially whether others are included in a factorial design? (L175-180). Please clarify

15- Why lithology is not included into the RegKrig? And why is topography included in ordKrig? Isn't ordKrig only based on spatial coordinates? Please clarify.

16- KGE is a very interesting metric. However, since high values represent better performance (contrary to NRMSE and deltaCORMAT) I would either use –KGE for the figures or remind in figure captions that high values=better performance (e.g. fig4).

17- l233: for the imputation of the whole dataset the authors use m=50 imputations because it is recommend to choose a value closed to the missingness rate. Why this same rule was not applied for the simulations as well?

18- It would also be important to test the correlations between environmental variables, to see at which point it is interesting or not to use highly correlated variables in the imputation.

Supplementary information (SI) I would recommend to reduce this part, 32 pages of material is a lot. I would suggest to try to reduce it to the most important results.

19- Sentence L.106 of the appendix is misleading, its seems that the authors use MICE-RF in the paper.

20- Maybe it would be clearer to merge s3 and s4

21- Please also place all references at the end of the appendix and not just after each paragraph, this makes it difficult to read the (already long) document.

22- S4: I do not see how PMM performs better than the other methods. Fig s4 shows that there is almost no difference between methods and some are better than others for some traits. Fig s5 and s6 only seem to show that mice_PAS is worse than the others. And nothing is supported by statistical tests. I would suggest to revise these analyses or provide a more complete explanation of why PMM was used. Please also add in the caption what the traits are or provide the complete trait names in the figure itself.

*Results and discussion*

This section is not very clear and sometimes the results are simply a description of

the figures without any effort to synthesise what happens. The choice for the figures is also not consistent (see point 23). It is titled "results and discussion" but the discussion is almost absent. For instance, I missed some information on why some methods perform better than others or why some traits show better results. Please see some more detailed comments below.

23- Figures in the main text are a bit confusing. Fig2 is NRMSE and 4 methods, fig3 is deltaCoeff and 4 methods, fig4 is KGE, but this time with 11 methods, fig5 is deltaCoeff with 11 methods, fig6 is KGE with 7 methods and fig7 is deltaCoeff with 7 methods. This is quite inconsistent and we miss some comparisons (e.g. mean and smean). I would suggest to put together in a different form (e.g. as in figs10) and show all the metrics (NRMSE, KGE, deltaCoeff) for each comparison. Also, the results about distributions (Kolmogorov-Smirnov tests) are not performed for all methods and only shown in the appendix.

24- In the whole first paragraph (l237) I am not sure that the differences highlighted by the authors are actually significant differences. For instance at 10% I only see Mean being slightly worse than the other methods for LMA and WD, and OrdKrig for WD. OrdKrig also does not seems to perform better than others for most traits. Same remark for Fig3. Maybe a test looking at the effects of the method corrected by the trait identity would provide more general results.

25- L250: from what I see in fig s8 mice is not the method with lower NRMSE or deltaCormat: kNN is the best performing method except at around 50% of NAs, where OrdKrig seems to perform better. This seems to be the case also for KGE where kNN performs as good as the other methods (fig s7). However it performs bad when looking at distributions. I also do not see mice being better than kNN or OrdKrig in Fig.3 (as stated L.251). I would suggest revising this paragraph and carefully checking all results for similar problems.

26- L 266: the best performance of mice_ctsp is not really visible in fig s11. - Fig s10

is also not easily readable, I would suggest to jitter the points (geom_jitter in ggplot2). Fig s12 is better (but jittering would help as well).

27- Mice-ctsp is discussed and presented several times (l.266, 291, 297, 306). I would suggest to merge together information. Also the 3.3 sections seems redundant with the 3.2. Maybe showing together fig6 and 7 would save some redundancy.

28- paragraph L 306 does not belong to section 3.3, which is about comparing mean, mice and knn. A separate section would be more meaningful.

29- L328: "MICE informed by relevant ecological variables outperforms": this was not properly tested as no analyses for this are provided

*Minor comments*

- L165: information about kNN is missing

- L234: "per missing value" should be "per missing dataset"

- L190: "statistical evaluation", no stats are actually performed for the evaluation

- To facilitate the reading, please cite the exact figures of the supplementary and not only the section (e.g. do not refer to just to s5 or s6 but to fig.s7 or s10).

- All figures: please spell out the trait names or write it in the caption

*References*

Nakagawa, S. and Freckleton, R. P.2008. Missing inaction: the dangers of ignoring missing data, Trends in Ecology & Evolution, 23(11), 592–596

Stekhoven, D. J. and Bühlmann, P. 2012. MissForest—non-parametric missing value imputation for mixed-type data. Bioinformatics 28: 112–118

van Buuren, S. and Groothuis-Oudshoorn, K., 2011, mice: Multivariate Imputation by Chained Equations in R, Journal of Statistical Software, 45(3)

---

## Referee Comment (RC2) · Anonymous Referee #2 · 12 Jan 2018

The paper compares different imputation methods for trait databases not only regarding their accuracy in predicting plant traits and trait-trait relationships but also their ability to preserve trait distribution and multi-trait correlation structures. They use a plant trait database that has complete trait observations for 5 traits in 630 plots together with the auxiliary information and the data are georeferenced. Exhaustive test are done to compare the methods in using only species mean data, adding auxiliary data or geographical information with different levels of gaps and 4 different complementary evaluation method i.e. R2, NRMSE, KGE, differences in the correlation matrices. This I believe is done for the first time in such a thorough way. The study is very relevant for biogeographical and ecological studies that need to use trait databases, which are

usually not free of gaps. However, compared to the number of methods and test done, there is not enough explanation and discussion done. I would suggest a re-evaluation of the discussion part. For example try to explain why the selected traits behave differently in the results.

I would suggest to add information on the abbreviations of the methods, auxiliary information and traits for all Figures also the figures in the supplementary. The Figure captions must be self-explanatory.

The paper uses different names like environmental information, ecological information/variables, and auxiliary information/variables etc. for the same thing and this is confusing. I suggest to use only auxiliary information or variable throughout the paper and to be consistent, especially since the data listed in the as ecological information (L90) are not all considered ecological exactly.

\*\*ABSTRACT\*\*

I suggest that the authors summarize the results in the order of the questions mentioned at the end of the introduction.

L 27: the word globally is misleading.

\*\*INTRODUCTION\*\*

This part is well written.

L 42: It is not clear what is meant by "primary sources"?

L 91: why is regKrig not in the second test together with MICE and kNN?

L91: perhaps would be better to define what you mean by "optimum level" here- best set of auxiliary variables selected in step ii - and then use the phrase hereafter

\*\*METHODS\*\*

Chapter 2.3: Please use this part to explain "Mean" and Spmean" technique a bit more.

L121: Please write the full name for MCAR as well.

L173: what type of "Lithology" data is used? Please include it here.

L175-178: Why are some auxiliary information added in a factorial design and some sequentially? What about running a test for each trait independently for realizing the most important auxiliary information that explains their variability best and then using those for the MICE and kNN model (also regKrig)?

Chapter 2.5: I would suggest making clear which statistical method is used for evaluating accuracy, multi-trait correlation structure or bivariate trait relationships.

L232: "as the best method globally" – Why globally?

\*\*RESULTS AND DISCUSSION\*\*

This part is in general not well written. There are a lot of results that looking at them in the paper and its' supplementary brings up many questions but the authors did not discuss them well. It's mostly like a report of results and not much discussion and reasoning of the results. Please discuss the performance discrepancies of the methods for different traits, and for the different evaluation criteria (trait-trait correlation, trait distribution, etc.).

First paragraph of chapter 3.1: Fig. 2 and Fig. S7 shows different results for different traits. How do you interpret this? Please discuss.

L239, L246, L250: Instead of Supplementary S5, please indicate Fig. S number. Please do so for the rest of the paper as well.

L246 Please remove "however".

L279: "stand structure" was mentioned before (L25, L90, L146, etc.) as "forest structure". Please choose one and be consistent.

Chapter 3.2: Why the comparison of adding different levels of auxiliary information

was only used for MICE and not for kNN or RegKrig? In addition, please discuss why mice_ctsp was chosen while according to Fig. 4 mice_sc would be sufficient for Nmass, LMA and WD and mice_st for Hmax and only BL:AS needs mice_cstp (extra p).

L288: Please replace A with S in "Fig. A7 and A12".

L288-289: what about adding only species and forest structure to RegKrig? Climate and topography should be already accounted for using spatial information in RegKrig.

L292-296: Please discuss and reason the differences of the results for the different traits.

L305: "(data not shown here)". But why not adding a figure similar to Fig. S8, for the comparison of Δcormat and dataset NRMSE between Spmean, mice_ctsp, kNN_ctsp and RegKrig in the supplementary as well?

L307-315: I would suggest to make this part chapter 3.4 (related to "Imputing traits for the main forest species in Catalonia 2.6)

**IMPLICATIONS**

This part is well written. I would suggest to add also suggestion for improvement of data collections, e.g. trait collection should be better be accompanied by auxiliary information on coordinates, forest structure, etc.

L322: "Here we deal here" Please remove the extra "here".

L357: Please change "practices" to "practiced".

**SUPPLEMENTARY**

Please refer to the supplementary with their Fig. numbers and not chapter numbers. Please make sure that all supplementary figures are mentioned in the main text. Currently this is not the case.

Fig S7: trait mean (Mean) is missing in the plot.

**BGD**

Fig S2: Please change "palant" to "plant".

Fig S6: Please define "cor.matrix abs.error" in the figure and "correlation matrix error" in the caption. Do you mean $\Delta$cormat? Please be consistent.

---

## Author Comment (AC1) · 2 Mar 2018

Key
Comments have numbered as Reviewer_number#comment_number (e.g. R1#12).

*Italics: original comments by the reviewer*
Normal font: response
**Bold: changes in the manuscript.**

Citations without reference correspond to papers cited in the manuscript. New references are specified here.

*R1#1. This paper compares the performance of different imputation methods for a trait dataset. Although this topic has already been studied before, the authors bring three novelties: (i) they use a spatially-explicit dataset (ii) which includes intra-specific information, and (iii) they use of a set of measures to assess imputation performance in terms of multivariate trait structure. This type of analysis is interesting, in a time where gap-filling methods are used more and more often to impute trait datasets.*

- We would like to thank the reviewer for acknowledging the novelty and relevance of our study.

*R1#2. However, the paper would need some clarifications and better justifications, in particular for the choice of the methods.*

- Following the reviewer's suggestion, we have made some changes to better justify the methods employed. We have modified the section '2.3. Imputation methods' so that the first clear paragraph now delineates the main reasons for the selection of imputation methods: mean and Spmean as baseline, widely used approaches, kriging methods, which account for spatial structure of the data and kNN/MICE, which are designed to impute multivariate datasets. The paragraph now reads (section 2.3):

- **We compared imputation methods with different degrees of complexity. We used two simple approaches to provide baseline imputations: Mean imputation ('Mean') filled missing data using the overall mean value for each trait and species mean imputation ('Spmean') replaced missing values with trait means computed for each species. Because of the spatial nature of the dataset, we also tested two geostatistical approaches, ordinary kriging ('OrdKrig') and regression kriging ('RegKrig'). Lastly, we also used two methods designed to handle multivariate datasets: *k*-nearest neighbour imputation ('kNN') and MICE (Multivariate Imputation using Chained Equations).**

- We have also improved the justification for some more specific methodological issues, such as the use of PMM or the specific tuning of kNN and MICE (see replies to comments *R1#9* , *R1#29*).

*R1#3. Some proper synthesis of the results is missing (shown by the total number of figures in main text and appendix) and no tests are performed, giving the general impression that none of the methods performs better than others.*

- We are aware that, although the main manuscript is not excessively long, the overall number of figures is very high. We deal with various different aspects related to the imputation of plant traits (imputation methods, auxiliary information used in the imputations, missingness, multiple traits) and these require extensive analyses. We also wanted to show the impact of different settings on MICE and kNN, because we often see that these methodological details are neglected in many studies, and one wonders whether they may cause differences in the final imputations.

- It is also true that, sometimes, we repeat (albeit expanded) some information in the Supplement that has already been shown in the main text (e.g. Fig. 4 and Fig. S10). Therefore, following the reviewer's comment, we propose to delete the following figures from the Supplement: Figs: S2, S4, S7, S10, S12, S13-S17. We will also merge sections S3 and S4 and put all references at the end of the supplement, following other comments by the reviewer (*R1#27, R1#28).*

- The reviewer comments that we do not perform statistical tests for all the results we show in the paper. Certainly, we do not use statistical tests for every single comparison, but we do use statistical inference in the last comparison (Supplement S8), which is, in our view, the most important. To make this more visible, we propose to move current Table S2 to the main text, as Table 1, showing how the different imputation methods perform with increasing missingness. This also implies a modification of the corresponding text in the results (section 3.3):

- **The LME model showed that the rate of increase in KGE with increasing missingness was lowest for 'Spmean' in four out of five traits (Table 1). Compared to 'Spmean' and 'RegKrig', performance of MICE and kNN declined more with increasing missingness (Table 1, Fig. S22, Table S1), but MICE generally outperformed kNN (Fig. 2, 3), as already observed in a recent imputation assessment of species-level, life-history traits (Penone et al. 2014).**

- In a revised version, we will propose to use mixed-effects models and pairwise comparisons to statistically test differences across methods. We will add the significance of the most important tests in the main text when describing the corresponding results (as would be described in a modified paragraph in the methods section 2.5):

- **Throughout the paper, we show violin plots representing the median and the distribution of each performance metric as a function of missingness levels, but we only graphically display the 10%, 30%, 50% and 80% levels, for ease of visualisation. We modelled imputation metrics in a linear mixed-effects model (LME) as a function of the interaction between imputation method and missingness, with dataset replicate as random effect. The LME model was fitted using the nlme package in R (Pinheiro et al., 2012) and pairwise comparisons of model coefficients were performed using the lsmeans and lstrends functions in the lsmeans package (Lenth, 2016).**

- We could also add the results of these multiple comparisons derived from the models in the main figures of the paper, except for Figure 4, where it would result in a highly cluttered figure. We would also restrict these for missingness of 50% and 80%, also to avoid unnecessary clutter. Although we don't think this would be necessary, given the changes we propose in the text (see above and comments to *R1#32, R1#33*), here is an example for Figure 2:

[Figure]

- See also some specific examples in some of our replies to your comments below (*R1#32, R1#33*).

*R1#4. There is also a number of problems in the results and figure presentations that should be fixed. Please refer to the comments below for more detail, in particular my points 7 to 10 are quite major.*

- We propose several changes to improve the presentation of figures and results. See also our responses to points 7-10 below (*R1#13 to R1#16*).

*R1#5. Another major point is the use of "ecological information" in the imputation process. The authors show that adding this information to the imputation improves its quality, as one would expect. However, in most ecological papers, authors usually look at the relationships between traits and this "ecological information". This introduces a serious problem of*

*circularity on any analysis using trait data imputed with "ecological information". The authors never mention this potential bias, do not suggest in which type of analyses such dataset could be used and more importantly, they do not test for this potential bias. A simple test would be to look at the relationship between traits and "ecological variables" in the complete dataset, in the datasets with missing data and in the imputed ones. This would give an idea about the importance of the bias and warn (or not, depending on the results) the users about it.*

- This is a reasonable concern, but the entire multiple imputation framework would tend to minimise the potential effects of circularity in an eventual analysis. The developers of MICE (van Buuren 2012, p.128; van Buuren & Groothuis-Oudshoorn, 2011, p. 22) in fact recommend to include in the imputation models those variables that will also be employed in the analysis of scientific interest (i.e. the trait or trait-environment relationship) and suggest that failure to do so would lead to biased results. This is because imputations are non-deterministic (i.e. they are drawn from a distribution) and the differences between multiply-imputed values represent the uncertainty in the imputation process. The multiple imputation framework provides tools to perform analyses on the multiply-imputed datasets, which include the uncertainty derived from the imputation process. Given the already complex and relatively long manuscript, we would prefer not to expand on this aspects, but we have modified a paragraph in the 'Implications' section 4 to deal explicitly with the circularity issue:

- **Compared to other imputation approaches, MICE is well-suited to deal with multivariate missing data (i.e. MICE produce imputations when some predictors are also missing) and provides information to quantify the uncertainty associated with the imputed data (Fig. 8). MICE uses multivariate relationships in the dataset to impute missing data, and this may raise concerns about potential circularity in analyzing trait-trait or trait-environment associations with the imputed dataset. Despite these concerns, multiple imputation practitioners argue that the full inference framework based on multiply-imputed datasets, would minimise the problem of circularity. Because our comparative assessment of imputation methods is already complex, here we have only dealt with *imputation*, the first step of the full process (e.g. Nakagawa & Freckleton 2008). MICE produces multiple datasets, with imputed values drawn from distributions, and these datasets can be combined in the *analysis* and *pooling* steps. The analysis step refers to the estimation of the parameters of scientific interest (e.g. a regression coefficient) for each dataset. In MICE, parameters can be pooled across datasets to produce unbiased estimates and standard errors, providing a natural way to take into account the additional uncertainty introduced in the analysis by the presence of missing data, and to minimize circularity issues (van Buuren 2012). However, ecological studies using multiple imputation approaches usually only apply the imputation step (Baraloto et al. 2010, Paine et al. 2011, Pyšek et al 2015, Díaz et al. 2016) and do not take advantage of the multiple imputation framework to quantify the uncertainty resulting from the presence of missing data (but see Fisher et al. 2003).**

*\*\*Abstract\*\**

*R1#6 (1)- L.16: "functional biogeography" is not mentioned elsewhere in the paper*

- 'Functional biogeography' appears in the 'implications' section.

*R1#7. (2)- L.16: "they offer specific challenges in terms of data imputation": these are not mentioned in the paper neither, it would actually be interesting to discuss these specific challenges*

- These aspects have been discussed in the paper, although the reviewer is correct that they could have been made more visible. For example, one important aspect is the fact that these are multivariate datasets and that they may also include missing data in the predictors. Another specific challenges are the spatial structure of the data (treated in our kriging approaches) and the fact that trait covariation and trait-environment relationships can be used to improve trait imputations (which we also address in the paper). We have added a paragraph in the 'Implications' section 4 discussing these issues:

- **This simulation study assesses different imputation methods in spatial, traits datasets with multivariate missing data. Amongst the methods assessed here, MICE and kNN are the most adequate to impute multivariate datasets, as they can be used when predictors also include missing data. Kriging methods may be more difficult to apply when predictors are also missing, but we have shown that, at high missingness levels and when environmental information is lacking, they can outperform MICE and kNN. This implies that methods including spatial variability may sometimes more provide more accurate imputations than those using trait covariation.**

*\*Introduction\**

*Overall, the introduction is well written and clear, it just lacks some details on some aspects (see below).*

*R1#8. (3)- Please provide a reference and explanation for the following statement L.56: "they all alter, to different degrees, the univariate trait distributions and the covariance structure of the dataset". Please also explain why multiple imputations do a better job in conserving trait distributions and covariances.*

- We acknowledge that our message here was a bit confusing. We now highlight that methods based on kNN and machine-learning are suitable to impute multivariate datasets and to preserve the covariance structure. We refer to Eskelson et al. 2009 and Penone et al. 2014 for examples on both types of methods. The modified sentences now read (section 1):

- **Single imputation methods replace a missing datum by one value and proceed with the analysis as if the imputed data had been observed (Nakagawa & Freckleton, 2008). Within these approaches, species mean or median imputation are probably the most widely used methods in ecology, but they ignore the variance of the imputed variables. Model-based imputation methods use other variables in the dataset to impute missing data, but they substantially alter the univariate trait distributions and the covariance structure of the dataset (Gelman & Hill, 2007). Approaches such as *k*-nearest neighbour (kNN) or machine-learning methods (Stekhoven & Bühlmann, 2012) may be**

**more appropriate to impute multivariate datasets, preserving their covariance structure (Eskelson et al. 2009; Penone et al. 2014).**

*R1#9. (4)- L60: Please cite which multiple imputation (MI) techniques are better to preserve structure and distribution of traits. MICE includes a high number of different algorithms, some accounting for variables distribution, others for interactions between variables, etc.. And actually some MICE techniques perform even worse than single imputations (e.g. see Stekhoven & Buhlmann 2012). MICE alone does not say much about the method and it would be important to specify quite early in the paper that it is MICEPMM which is tested.*

- We agree with the reviewer that MICE includes several variants, mainly defined by the specific univariate imputation model. In fact, our study is the first one to our knowledge to make a comprehensive assessment of different variants within MICE within the biological/ecological literature. Other studies have either used PMM directly (Penone et al 2014) or have not even reported which univariate method was used within MICE (Stekhoven & Buhlmann 2012). We wondered whether the choice of method could have influenced the results of these papers, and it appears that it has not (see below).

- As for the information on the univariate imputation method in MICE early in the paper, we prefer not to include this in the introduction, and leave the description and justification of the use of PMM in the Methods section. The aim of our study is broader and includes multiple aspects (imputation methods, auxiliary information used in the imputations, missingness, multiple traits) and we prefer not to include such technical details in the introduction. As we state in the Methods, we chose PMM because it is known to perform better when non-normal distributions and non-linear relationships between variables are present (see Morris et al. 2014, now cited in the main text). But we also tested its performance compared to other algorithms (included a random-forest based one), and while it is true that in our tests PMM did not show a big improvement over the other algorithms, it did not perform worse either.

*R1#10. (5)- L 62. I would not define kNN as a sophisticated method, it's in fact quite a simple One*

- We have replaced 'sophisticated' by 'statistical'. The text now reads (section 1):
- **While forest inventories have adopted statistical imputation methods for some time, as for example the kNN methods (Eskelson et al. 2009 and references therein), imputation methods have only recently been started to be used in trait-based ecology (Baraloto et al. 2010; Pyšek et al. 2015).**

*R1#11. (6)- L90: sampling date is not really "ecological information", also most of the predictors mentioned were not really introduced before.*
- We agree with the reviewer that the term 'ecological information' may not include all the different variables proposed here. We will use 'environmental information

throughout the paper instead of 'ecological information'. As for the predictors mentioned here, they were genericaly introduced previously in the introduction, but without mentioning them specifically (section 1):

- **However, intraspecific variability in plant traits may be substantial (Siefert et al. 2015; Vilà-Cabrera et al. 2015) and imputation methods that use environmental information may be more appropriate when assessing trait relationships and trait-environment covariance in a spatially explicit context. Biotic or abiotic variables other than the trait matrix of interest can be included in imputation algorithms as auxiliary variables to reduce imputation bias (Azur et al. 2011; Rezvan et al. 2015).**

*Methods*

*R1#12. In general, the methods are clearly explained but some things are understandable only when reading the supplementary information. The methods are quite dense, so it is OK to have some descriptions in the appendix. However, they should be selfunderstandable. Figure 1 is very helpful and important. I also think that the evaluation metrics are very interesting. However, I did not understand some of the choices, which I think should be better justified (see below).*

- We would like to thank the reviewer for the positive comments on our methodological description, including the praise for Fig. 1 and the choice of metrics. We expect to clarify some of the issues on clarity and justification below.

*R1#13. (7)- A quite major point is that the authors state that in the real dataset missing data is biased towards leaf traits deliberately. So data are not MCAR in the dataset (l.120). We also know that in trait datasets, values are often MAR (Nakagawa & Freckleton 2008). But the authors then remove data completely at random (L121). We also know that imputation methods are not designed for MCAR data (van Buuren & GroothuisOudshoorn, 2011). So why the authors chose to remove data MCAR? I would suggest to introduce data at random with the same structure as in the original dataset. This would be a fairer test than just removing data completely at random. It is important to*
*see how the methods behave when data is MAR.*

- As correctly pointed out by the reviewer, missing data in foliar traits were introduced deliberately and randomly, following the sampling design, because these traits were measured only in a random subset of the plots. Intentional (van Buuren 2012) or planned missing data design (Nakagawa 2015) are implemented when a certain variable, or set of variables, is costly to measure but another set of variables, the auxiliary variables, can be used in the imputations. And this is considered in our approach of including auxiliary information to improve imputation performance (for example, inclusion of topography would take into account any geographical pattern in missingness levels).

- Therefore, for foliar traits, a large proportion of the missing data were actually MCAR by design. However, we agree with the reviewer that, in general, missing data in our dataset is probably MAR. MAR implies that the probability of missingness depends on another variable in the dataset. As a simple test, we fitted the probability of

missingness for all traits in the IEFC incomplete dataset using a logistic model including species as a factor. For all traits, except Hmax, species had a significant effect, but there was not a clear pattern of missingness associated to certain taxonomic groups (e.g. higher missingness in Pinaceae vs Fagaceae). This shows that there is no consistent pattern in the missing data mechanism within or across traits.

- Hence, we agree with the reviewer that simulations using MAR datasets may have been more realistic. However, because of the results mentioned above, but it would have been difficult to decide on how to apply this MAR assumption when introducing gaps in the datasets. In other studies missingness has been more consistently related to a variable in the dataset, allowing for an easier application of the MAR assumption. For example, Penone et al. 2014 simulated MAR gaps by assuming that missingness was higher in small carnivores, compared to large ones, according to missing data patterns observed in trait datasets. In that study, they did not find any difference compared to simulations using MCAR missing data.

- Given these prior results and the approaches we follow towards the inclusion of auxiliary variables in the imputations, we did not expect a major impact of different missing data mechanisms in our simulations. In addition, our study already deals with many aspects (imputation methods, auxiliary information used in the imputations, missingness, multiple traits) and adding yet another source of variability would likely make the paper overly complex. However, we have added a sentence in the 'Implications' section recognising that there is some uncertainty in the results that arises from the assumption of missing data being MCAR (section 4):
- **One limitation of this study is that we simulate MCAR missing data when a MAR assumption would have probably been more realistic (Nakagawa 2015), although a recent study did not show differences in trait imputation performance between these two missing data mechanisms (Penone et al. 2014).**

*R1#14 (8)- Another point is that there is no comparison with the dataset including missing data. The question: "should I impute or not" is an important one so it would be good to know how well the imputation methods perform (in terms of KGE and structure) when compared to just using the dataset with gaps.*

- We indeed compared results (for the differences in trait relationships) when using complete, incomplete or imputed datasets (Figs. S20, S21). We have now expanded this part of the results (section 3.3):
- **Using incomplete (i.e. not imputed) datasets to retrieve trait correlations had an impact on the Nmass−LMA relationship at high missingness rates, compared to 'mice_ctsp' imputations (i.e. larger departure from the 'complete' line for 'incomplete' compared to 'imputed', Fig. S20), but this trend was not clear for Hmax −WD (Fig. S21). Many imputation methods did not seem to improve analyses performed with the incomplete dataset (Fig. S21). For example, kriging and 'Spmean' methods showed the largest departures in trait**

**relationships, when compared against analyses performed with 'complete' or 'incomplete' datasets (Fig. S21, S22).**

- We would also like to stress the fact that in some cases, imputation is really necessary, because some representation of geographic variability is needed for applications such as trait mapping or trait-driven modelling approaches. We now highlight this in the closing statement of the manuscript (section 4):

- **We also show that spatially-distributed layers of environmental information may improve trait mapping, increasing spatial resolution and/or sample size in trait-driven ecosystem process models (Christoffersen et al. 2016).**

*R1#15 (9)- Related to the previous, at the end, users would like to know which method is the best, considering together NRME, distributions, correlations, structure and regardless of the trait considered or the % of missing data. This could be analysed using a mixed model with the identity of the metric (NRMSE, KGE..etc) and the missing dataset identity as random factors.*

- We have shown that it is difficult to provide an unequivocal response to the question of what methods performs best in all situations. We show that, when no auxiliary variables are present, using a geostatistical method such as ordinary kriging can significantly improve imputations, compared to simple averaging approaches, but also compared to approaches using trait covariation (MICE, kNN). We show that, at intermediate missingness rates, MICE with some auxiliary variables tends to be the best performing method (especially if we consider distributions and the metrics related to covariance structure) but this advantage vanishes at higher missingness rates, and the best-performing method depends a lot on which trait we consider.

- Given these results, we emphasise the usefulness of MICE because it is naturally designed to handle multivariate missing data (mixed data types, missing data in the predictors) and it provides uncertainty estimations around the imputations. For example see our text in the 'Implications' section 4:

- **Compared to other imputation approaches, MICE is well-suited to deal with multivariate missing data (i.e. MICE produce imputations when some predictors are also missing) and provides information to quantify the uncertainty associated with the imputed data (Fig. 8).**

*R1#16 (10)- Given that the dataset has intraspecific variation and this is presented as a novelty both in the introduction and the "implication" section, I would have expected an analysis of the error at the species level. Also, it should be noted that adding species identity assumes that interspecific variation is higher than intraspecific variation, which is OK, but should maybe be stated somewhere.*

- Our study focused on questions that are of more general interest to ecologists (imputation methods, use of auxiliary variables) rather emphasising results that would be more specific to the study system in question (species-specific patterns). Moreover, a recent study (Vilà-Cabrera et al., 2015) has already looked at trait variability at different taxonomic levels, including the influence of environmental gradients on the same dataset.

- In our factorial experimental design we have seen that species identity always improved imputations, but for some cases (Hmax, for example) it is not the single

predictor that contributes most to improve the imputations (in this case it's forest structure). This factorial design, therefore, does not assume a priori higher interspecific variation and shows that other auxiliary variables may be better predictors than species identity.

*R1#17. (11)- L. 126: twhy m=5? MICE authors recommend to use at least 10 iterations (van Buuren& Groothuis-Oudshoorn, 2011). I saw later that this was explained in the appendix, I think that it should be at least mentioned in the main text.*

- The number of iterations (t) refers to the number of cycles through all missing variables in the dataset, and we set that to 20, more than the default number of iterations in the MICE algorithm (t = 5). We do this to improve the stabilisation of parameters in the imputation model and to minimise the effect of imputation order. We explain this in the supplementary material S4.
- The number of multiply imputed datasets (m) was set to 5, the default in the MICE algorithm, because this is a recommended setting during the test stage (van Buuren 2012) and because it makes simulations more efficient computationally (in terms of both computing time and data generation). This is also explained in detail in the supplementary material S4.

*R1#18 (12)- L.138: similar to the previous point, it would be good to state that k=7 comes from*
*an additional analysis (described in the appendix), otherwise the "7" is quite obscure.*
*All these omissions make the reading/methods understanding quite challenging.*

- We have rewritten this part of the Methods and now we explicitly say that these methodological decisions are a result of the preliminary tests performed:
- **We selected *k* = 7 and median aggregation after some preliminary tests (Supplement S2).**

*R1#19. (13)- The MICE paragraph (l.151) should give more details on what is described in appendix S4.*
- We decided to include the detailed explanation of the MICE algorithm in the Supplement because otherwise the Methods section would be even longer than it is now. We think that this makes the paper shorter and easier to read. What we have done is to merge sections S3 and S4 in the Supplement, so now the MICE explanation is more compact (see *R1#27*).

*R1#20. (14)- Why some variables are added sequentially whether others are included in a factorial design? (L175-180). Please clarify*

- We were interested in identifying which combinations of the variables with a major role in explaining trait variability (species identity, climate and forest structure; see Vilà-Cabrera et al 2015), led to improved imputations. Other variables which we expected to play a secondary role (topography, lithology, sampling month) were

added sequentially. We have rewritten the corresponding explanation in the methods section, to make this clear  (section 2.4):

- **Species identity ('s'), climate ('c') and forest structure ('t') were introduced in a factorial design to identify those combinations of variables leading to improved imputations. Because we expected them to play a secondary role in explaining trait variability, topography ('p'), lithology ('l') and sampling month '(m)' were sequentially added to MICE and kNN imputations using species, climate and forest structure.**

*R1#21. (15)- Why lithology is not included into the RegKrig? And why is topography included in ordKrig? Isn't ordKrig only based on spatial coordinates? Please clarify.*

- As noted by the reviewer, ordinary kriging is only based on the geographic coordinates, as explained in the beginning of section 2.4. Lithology (as topography, or sampling month) was considered a predictor of secondary importance, compared to species identity, climate and forest structure. For this reason, and to avoid increasing the complexity of the analysis, these variables were only investigated within the mice imputations and not used in the other methods (kNN, kriging).

*R1#22. (16)- KGE is a very interesting metric. However, since high values represent better performance (contrary to NRMSE and deltaCORMAT) I would either use –KGE for the figures or remind in figure captions that high values=better performance (e.g. fig4).*

- Following the reviewer's suggestion, we will add a note in the caption explaining that high values of KGE mean better performance.

*R1#23. (17)- l233: for the imputation of the whole dataset the authors use m=50 imputations because it is recommend to choose a value closed to the missingness rate. Why this same rule was not applied for the simulations as well?*

- As also mentioned in comment #11 above, we followed the recommendations by the developers of MICE and we used 5 imputations during the test stage and a larger number for the application stage (van Buuren 2012). This avoided a large computational burden, in terms of processing time and data handling/storage.

*R1#24. (18)- It would also be important to test the correlations between environmental variables, to see at which point it is interesting or not to use highly correlated variables in the Imputation.*

- This is a good point, because it is known that MICE may have problems when input variables are highly correlated. We minimised this by only selecting variables which already showed low correlation (r<0.3) in a previous study using the same dataset (Vilà-Cabrera et al. 2015). In addition, we performed some preliminary tests using the 'quickpred' function included in the 'mice' package, which allows to select only those predictors below a given correlation value. These tests showed no improved performance compared to our 'mice' settings.

*R1#25. Supplementary information (SI) I would recommend to reduce this part, 32 pages of material is a lot. I would suggest to try to reduce it to the most important results.*

- We agree with the reviewer that the supplementary materials are very long. We have substantially reduced the supplementary materials (see commenr R1#3 for more details) and now, instead of 32 pages, they are 19 pages long.

*R1#26 (19)- Sentence L.106 of the appendix is misleading, its seems that the authors use MICE-RF in the paper.*

- We have rewritten the sentence to make it clear that we refer to the comparison of univariate imputation models within mice:
- **Here, in this comparison of univariate imputation models, we used the implementation of the random forest algorithm in MICE, a described in Doove *et al.*, (2014).**

*R1#27 (20)- Maybe it would be clearer to merge s3 and s4*

- Following the reviewer's suggestion, we have merged S3 and S4.

*R1#28 (21)- Please also place all references at the end of the appendix and not just after each paragraph, this makes it difficult to read the (already long) document.*

- Done.

*R1#29 (22)- S4: I do not see how PMM performs better than the other methods. Fig s4 shows that there is almost no difference between methods and some are better than others for some traits. Fig s5 and s6 only seem to show that mice_PAS is worse than the others. And nothing is supported by statistical tests. I would suggest to revise these analyses or provide a more complete explanation of why PMM was used. Please also add in the caption what the traits are or provide the complete trait names in the figure Itself.*

- We have already discussed the choice of PMM in an earlier comment (R1#9), as a method which can handle non-linear relationships and non-normality in the data. We think that a thorough analysis of the justification of PMM is too technical for the scope of this manuscript, and our preliminary tests may be viewed as a corroboration that PMM did not perform worse than other more computationally-intensive methods such as RF (mice_rf).
- We have added a full description of the traits in the corresponding figures. An example:
- **Figure S3. Trait-specific imputation performance (NRMSE) at increasing missingness levels (10% to 80%) using different MICE settings (see text for details). The following approaches used the predictive mean matching method (PMM) as the univariate imputation model: passive imputation of derived variables (mice_PAS), derived variables imputed as 'just another variable' (mice_JAV), imputation using log-transformed variables (mice_TRN). mice_PRD differed from PMM in that mice_PRD used the predicted trait from the sequential, multiple regression models in the MICE framework, without the**

**stochasticity in regression coefficients that is introduced in PMM. mice_RF uses a random-forest algorithm for univariate imputation instead of PMM (see text). The original trait distributions in the IEFC complete data set (Observed) are also shown. Traits: leaf biomass to sapwood area ratio, BL:AS (t m-2); nitrogen per unit mass, Nmass (%mass); maximum tree height, Hmax (m); leaf mass per area LMA (mg cm-2); wood density, WD, (gm cm-3).**

*\*Results and discussion\**
*R1#30. This section is not very clear and sometimes the results are simply a description of the figures without any effort to synthesise what happens. The choice for the figures is also not consistent (see point 23). It is titled "results and discussion" but the discussion is almost absent. For instance, I missed some information on why some methods perform better than others or why some traits show better results. Please see some more detailed comments below.*

- We have changed the 'Results and discussion' section extensively, adding more statistical tests and rearranging parts of the text. See below for more detailed explanations.

*R1#31 (23)- Figures in the main text are a bit confusing. Fig2 is NRMSE and 4 methods, fig3 is deltaCoeff and 4 methods, fig4 is KGE, but this time with 11 methods, fig5 is deltaCoeff with 11 methods, fig6 is KGE with 7 methods and fig7 is deltaCoeff with 7 methods. This is quite inconsistent and we miss some comparisons (e.g. mean and smean). I would suggest to put together in a different form (e.g. as in figs10) and show all the metrics (NRMSE, KGE, deltaCoeff) for each comparison. Also, the results about distributions (Kolmogorov-Smirnov tests) are not performed for all methods and only shown in the appendix.*

- There are some reasons behind the differences in the figure design. For the first comparison, KGE is not defined for 'Mean' imputation, and therefore we chose to represent NRMSE (but KGE can be found in the supplement for those methods for which it can be calculated). The figures show different number of methods because they reflect different comparisons, as outlined in the 'Methods' section. We agree with the reviewer that some comparisons cannot be made directly, but we had to choose one way of presenting the results and this way reflects the questions that we posed in the introduction. We also chose the violin plots in the main text to better compare distributions of the different metrics across imputation methods at a given missingness level, and show a different visualisation in the supplement to focus on the trends with missingness.

*R1#32 (24)- In the whole first paragraph (l237) I am not sure that the differences highlighted by the authors are actually significant differences. For instance at 10% I only see Mean being slightly worse than the other methods for LMA and WD, and OrdKrig for WD. OrdKrig also does not seems to perform better than others for most traits. Same remark for Fig3. Maybe a test looking at the effects of the method corrected by the trait identity would provide more general results.*

- As explained before (*R1#3*), we have now performed pairwise comparisons resulting from a model including imputation method and missingness as predictors. We have also modified the first paragraph of the 'Results and discussion' section using this new approach. In this paragraph we now emphasize more the differences between traits and convey the message that, without auxiliary variables, 'OrdKrig' performs better that the rest of the methods (in terms of NRMSE, see section 3.1):

- **In general, 'mice' and 'kNN' imputations resulted in more accurate imputations in terms of NRMSE than 'Mean' at low missingness rates (10%). However, at moderate and high missingness both 'mice' and 'kNN' were comparable or outperformed by 'Mean', and specially by 'OrdKrig' (Fig. 2, Fig. S7). 'OrdKrig' was the best-performing method, in terms of NRMSE, at missingness ≥ 50% (P <0.05), although for three traits its performance was indistinguishable from that of 'Mean' imputations (Nmass, Hmax, LMA; P>0.05). Even if 'Mean' imputations imply the rather naive assumption that species identity may be unknown in a given dataset, it is nonetheless useful to compare 'Mean' imputations against 'mice' and 'kNN', which use the full trait matrix for prediction. In this case, trait covariation did not improve imputations at high missingness. Recent assessments also report that the performance of MICE and kNN notably declines when missingness is ≥ 30% (Penone et al. 2014; Taugourdeau et al. 2014). Therefore, our results for 'OrdKrig', compared to those for 'mice' and 'kNN', show that spatial structure, rather than trait covariation, may provide more accurate trait imputations when gaps are frequent (Fig. 2, Fig. S7, S8).**

*R1#33 (25)- L250: from what I see in fig s8 mice is not the method with lower NRMSE or deltaCormat: kNN is the best performing method except at around 50% of NAs, where OrdKrig seems to perform better. This seems to be the case also for KGE where kNN performs as good as the other methods (fig s7). However it performs bad when looking at distributions. I also do not see mice being better than kNN or OrdKrig in Fig.3 (as stated L.251). I would suggest revising this paragraph and carefully checking all results for similar problems.*

- We do not state that mice has the lowest NRMSE or Δcormat. For example, we say that mice and kNN seem to perform better at low missingness but that their performance seems to decline with missingness (section 3.1):

- **In general, 'mice' and 'kNN' imputations resulted in more accurate imputations in terms of NRMSE than 'Mean' at low missingness rates (10%). However, at moderate and high missingness both 'mice' and 'kNN' were comparable or outperformed by 'Mean', and specially by 'OrdKrig' (Fig. 2, Fig. S7). [...]**

- The reviewer points out at the Δcormat performance of kNN at 50%, which we also mention (section 3.1):
- **'Mean' imputation severely altered trait distributions (Fig. S9), and introduced larger errors in selected trait correlations (Fig. 3). 'Mean' imputations also tended to cause larger deviations in the correlation matrix (Fig. S8). 'kNN'**

**showed the lowest Δcormat below 50% missingness (P>0.05) but its performance declined at high missingness (Fig. S8).**

- Moreover, as the reviewer comments,  we already distinguish in the text between the performance, in terms of accuracy and in terms of preserving relationships and multivariate structure (section 3.1).
- **In contrast, 'mice' closely tracked observed trait distributions (Fig. S9), introduced the least error in trait correlations under high missingness levels (Fig. 3; P<0.05) and yielded low Δcormat at extreme missingness levels (Fig. S8).**

- We have rewritten the comparison of Δcormat between 'mice' and 'OrdKrig' to better reflect the fact they tend show similar results,as noted by the reviewer (section 3.1).
- **'OrdKrig' imputations altered distributions and trait correlations more than 'mice' (Fig. 3, Fig. S9), but they performed similarly in terms of Δcormat at missingness of 80% (Fig. S8).**

*R1#34 (26)- L 266: the best performance of mice_ctsp is not really visible in fig s11. - Fig s10 is also not easily readable, I would suggest to jitter the points (geom_jitter in ggplot2). Fig s12 is better (but jittering would help as well).*

- As we explain in the reply to comment *R1#3,*  and  in line with  proposed reductions in the length of supplementary materials, we actually propose to delete Fig. S10 and Fig. S12 because they're partially redundant (results are already shown in Fig. 4 and Fig. 6). As for Fig. S11, we are aware that differences can not be seen, but the point of the figure is precisely that distributions are very similar across MICE applications.

*R1#35 (27)- Mice-ctsp is discussed and presented several times (l.266, 291, 297, 306). I would suggest to merge together information. Also the 3.3 sections seems redundant with the 3.2. Maybe showing together fig6 and 7 would save some redundancy.*

- We think that it would be difficult to merge Figs. 6 and 7, because the resulting figure would have too many panels and would be difficult to read. About the section 3.3., we have made extensive changes to the entire 'Results and Discussion' section towards a better clarity.
- As for the repeated appearances of 'mice_ctsp', this is because the comparisons in the second and third objective both include this imputation method. In the second objective we compare it against MICE applications with different levels of auxiliary variables and in the third objective we compare it against other imputation methods.

*R1#36(28)- paragraph L 306 does not belong to section 3.3, which is about comparing mean, mice and knn. A separate section would be more meaningful.*

- Done. We have added a new section header: 3.4. Imputing traits for the main forest species in Catalonia.

*R1#37(29)- L328: "MICE informed by relevant ecological variables outperforms": this was not properly tested as no analyses for this are provided*

- We now provide statistical tests for the most meaningful comparisons (see replies in *R1#32, R1#33.*

*Minor comments*

*R1#38 - L165: information about kNN is missing*

- Thanks for spotting this mistake. We have rewritten the sentence (section 2.4):

- **'Mean' imputations used only the information on the target trait, 'OrdKrig' additionally used the spatial coordinates and 'mice' and 'kNN' included only the information in the trait matrix.**

*R1#39 - L234: "per missing value" should be "per missing dataset"*

- The text is actually right because in multiple imputation, >1 values are generated for each missing value.

*R1#40- L190: "statistical evaluation", no stats are actually performed for the evaluation*
- We now provide statistical tests for the most meaningful comparisons (see examples in *R1#32, R1#33*)

*R1#41 - To facilitate the reading, please cite the exact figures of the supplementary and not only the section (e.g. do not refer to just to s5 or s6 but to fig.s7 or s10).*
- Done.
*R1#42 - All figures: please spell out the trait names or write it in the caption*
- Done in the caption.See an example in our reply to comment *R1#29.*

*References*
*Nakagawa, S. and Freckleton, R. P.2008. Missing inaction: the dangers of ignoring missing data, Trends in Ecology & Evolution, 23(11), 592–596*
*Stekhoven, D. J. and Bühlmann, P. 2012. MissForestâA ̆Tnon-parametric missing value ̆ imputation for mixed-type data. Bioinformatics 28: 112–118*
*van Buuren, S. and Groothuis-Oudshoorn, K., 2011, mice: Multivariate Imputation by Chained Equations in R, Journal of Statistical Software, 45(3)*

---

## Author Comment (AC2) · 2 Mar 2018

Key
Comments have numbered as Reviewer_number#comment_number (e.g. R2#12).

*Italics: original comments by the reviewer*
Normal font: response
**Bold: changes in the manuscript.**

Citations without reference correspond to papers cited in the manuscript. New references are specified here.

*R2#1. The paper compares different imputation methods for trait databases not only regarding their accuracy in predicting plant traits and trait-trait relationships but also their ability to preserve trait distribution and multi-trait correlation structures. They use a plant trait database that has complete trait observations for 5 traits in 630 plots together with the auxiliary information and the data are georeferenced. Exhaustive test are done to compare the methods in using only species mean data, adding auxiliary data or geographical information with different levels of gaps and 4 different complementary evaluation method i.e. R2, NRMSE, KGE, differences in the correlation matrices. This I believe is done for the first time in such a thorough way. The study is very relevant for biogeographical and ecological studies that need to use trait databases, which are usually not free of gaps. However, compared to the number of methods and test done, there is not enough explanation and discussion done. I would suggest a re-evaluation of the discussion part. For example try to explain why the selected traits behave differently in the results.*

*R2#2. I would suggest to add information on the abbreviations of the methods, auxiliary information and traits for all Figures also the figures in the supplementary. The Figure captions must be self-explanatory.*
  - Done. Here is the caption for Figure 4 as an example:
  - ***Figure 4. Trait-specific KGE at increasing missingness levels (10% to 80%) and for different MICE imputations using different combinations of additional predictor sets: species identity (s), climate (c), forest structure (t), sptatial structure (p), lithology (l) and sampling month (m). See Fig. 1 for an overall view of the experimental design and the Methods section for a detailed description of the variables employed in each predictor set. Traits: leaf biomass to sapwood area ratio, BL:AS (t m-2); nitrogen per unit mass, Nmass (%mass); maximum tree height, Hmax (m); leaf mass per area LMA (mg cm-2); wood density, WD, (gm cm-3). Higher values of KGE imply higher performance.***

*R2#3. The paper uses different names like environmental information, ecological information/variables, and auxiliary information/variables etc. for the same thing and this is confusing. I suggest to use only auxiliary information or variable throughout the paper and to be consistent, especially since the data listed in the as ecological information (L90) are not all considered ecological exactly.*

*\*\*ABSTRACT\*\**

*R2#4. I suggest that the authors summarize the results in the order of the questions mentioned at the end of the introduction.*
  - The structure of this section of the abstract, where we outline the results, is as follows: (i) using averaging to impute trait datasets may work well, on average, but these methods distort trait variability and covariation, (ii) which environmental variables are more useful to produce

plausible imputations, (iii) how do results vary across traits and missingness levels and (iv) describe the application of a best-performing method for the entire dataset. We are aware that the structure suggested by the reviewer would make easier to map the results between different sections of the paper. However, during early stages of manuscript writing, and because of the highly technical nature of the paper, we decided to mix results and discussion. In the case of the abstract, the messages we try to convey are better expressed using this logic, rather than the order of the questions in the Introduction, which are more related to the three different 'exercises' in the manuscript. For these reasons, we prefer to keep the present Abstract structure.

*R2#5. L 27: the word globally is misleading.*

- We have replaced 'globally' by 'overall'.

*\*\*INTRODUCTION\*\**

*This part is well written.*
*R2#6 L 42: It is not clear what is meant by "primary sources"?*

- We refer to individual studies from the primary literature (mainly research papers). For clarity, we have removed 'primary sources and' so the text now reads (section 1):
- **Plant trait databases compiled from multiple individual contributions lack a common design and inevitably result in sparse data matrices (e.g. Jetz et al. 2016).**

*R2#7. L 91: why is regKrig not in the second test together with MICE and kNN?*

- The second exercise deals with how auxiliary variables can improve imputations, and we chose to do this analysis only for MICE and kNN. Both methods are easy to implement for multivariate datasets in our simulation framework, but regression kriging is not such a flexible tool. For example, we could not include species identity for regKrig because, for less common species, there were not enough data to perform the regKrig imputations at high missingness rates.

*R2#8. L91: perhaps would be better to define what you mean by "optimum level" here- best set of auxiliary variables selected in step ii - and then use the phrase hereafter*

- We have rewritten the sentence following the reviewer's suggestions (section 1):
- **(iii) to compare the performance of kNN, MICE and RegKrig using optimum levels of ecological information (i.e. the best set of predictors in objective ii);**

*\*\*METHODS\*\**

*R2#9. Chapter 2.3: Please use this part to explain "Mean" and Spmean" technique a bit more.*
- We think that 'Mean' and 'Spmean' are already well explained when we present the imputation methods (section 2.3):
- **We compared imputation methods with different degrees of complexity. We used two simple approaches to provide baseline imputations; Mean imputation ('Mean') filled missing data using the overall mean value for each trait and species mean imputation ('Spmean') replaced missing values with trait means computed for each species.**

*R2#10. L121: Please write the full name for MCAR as well.*
- We have rewritten the sentence following the reviewer's suggestion.

- **In this dataset, we randomly deleted measured values at different probability levels (10%, 20%, 30%, 50% and 80%) and independently for each trait, thus the missing data artificially introduced are missing completely at random (MCAR).**

*R2#11. L173: what type of "Lithology" data is used? Please include it here.*

- Lithology includes three categories: calcareous, non-calcareous and undetermined. We have added this information to the sentence (section 2.4).
- **The auxiliary variables we considered were species identity, a set of climatic variables (mean annual temperature, annual thermal amplitude, both in °C), a set of forest structure variables (total aboveground biomass [T ha⁻¹] and stem density [stems ha⁻¹]), a set of topographical variables (county, elevation [m.a.s.l.], slope [°] and aspect), lithology (calcareous, non-calcareous or undetermined) and sampling month.**

*R2#12. L175-178: Why are some auxiliary information added in a factorial design and some sequentially? What about running a test for each trait independently for realizing the most important auxiliary information that explains their variability best and then using those for the MICE and kNN model (also regKrig)?*

- We were interested in identifying which combinations of the variables with a major role in explaining trait variability (species identity, climate and forest structure; see Vilà-Cabrera et al 2015), led to improved imputations, and for this reason we included these variables in a factorial design. Other variables which we expected to play a secondary role (topography, lithology, sampling month) were added sequentially. See also response *R1#20 above.*
- We did this exercise mainly for MICE, but also somehow for kNN, because they were easier to implement in our simulation framework and they are both algorithms that are designed to deal with multivariate missing data (as opposed to regression kriging).
- We are not really sure to understand what the reviewer is suggesting here. Does the reviewer suggest to use different information levels depending on each trait instead of using a best-set (e.g. mice_ctsp) for all traits? We think that our approach deals with the role of auxiliary variables in imputation performance at two levels: (1) which combinations of relevant ecological variables improve imputations and (2) which secondary variables can help improve those imputations.

*R2#13. Chapter 2.5: I would suggest making clear which statistical method is used for evaluating accuracy, multi-trait correlation structure or bivariate trait relationships.*

- Following the reviewer's suggestion, we have added a clarification on the association of NRMSE as a measure of imputation accuracy (section 2.5):
- **For each simulated dataset and trait, we calculated the Normalised Root Mean Square Error (NRMSE) as a measure of accuracy**

- Also, following the reviewer's suggestion we have explicitly added 'multi-trait correlation structure' in the description of Δcormat (section 2.5):
- **The deviations from the original multi-trait correlation structure of the trait dataset were quantified by comparing the correlation matrices of the original and imputed datasets using the following index:**

- As for the impact on bivariate trait relationships, we consider that the explanation is already clear (section 2.5):
- **We also tested the impact of the imputation algorithms on selected bivariate trait relationships: $H_{max}$–WD and $N_{mass}$–LMA (log-transformed when necessary); as the**

**correlation coefficients (*r*) of these relationships were >0.3 in absolute value and were highly significant in the complete dataset. We quantified the relative difference between the complete and the imputed datasets by calculating:**

- KGE is a more complex indicator, which also includes accuracy, and we think that its current explanation is clear enough  (section 2.5):
- **KGE jointly assesses correlation, bias and variability between imputed and observed values, and it is therefore a powerful, synthetic indicator of imputation quality in spatially-explicit datasets.**

*R2#14. L232: "as the best method globally" – Why globally?*

- We mean 'overall', considering all traits and performance metrics, as explained later in the sentence. We have replaced 'globally' by 'overall'.

*\*\*RESULTS AND DISCUSSION\*\**

*R2#15. This part is in general not well written. There are a lot of results that looking at them in the paper and its' supplementary brings up many questions but the authors did not discuss them well. It's mostly like a report of results and not much discussion and reasoning of the results. Please discuss the performance discrepancies of the methods for different traits, and for the different evaluation criteria (trait-trait correlation, trait distribution, etc.).*

- We have changed the 'Results and discussion' section extensively, adding more statistical tests and rearranging parts of the text. See our replies to comments *R2#16, R2#21 and R2#24.*

*R2#16. First paragraph of chapter 3.1: Fig. 2 and Fig. S7 shows different results for different traits. How do you interpret this? Please discuss.*
- We have now rewritten this entire paragraph and we place more emphasis on particular results of some traits (Nmass, Hmax, LMA, see section 3.1).
- **In general, 'mice' and 'kNN' imputations resulted in more accurate imputations in terms of NRMSE than 'Mean' at low missingness rates (10%). However, at moderate and high missingness both 'mice' and 'kNN' were comparable or outperformed by 'Mean', and specially by 'OrdKrig' (Fig. 2, Fig. S7). 'OrdKrig' was the best-performing method, in terms of NRMSE, at missingness ≥ 50%  (P <0.05), although for three traits its performance was indistinguishable from that of 'Mean' imputations (Nmass, Hmax, LMA; P>0.05).  Even if 'Mean' imputations imply the rather naive assumption that species identity may be unknown in a given dataset, it is nonetheless useful to compare 'Mean' imputations against 'mice' and 'kNN', which use the full trait matrix for prediction. In this case, trait covariation did not improve imputations at high missingness. Recent assessments also report that the performance of MICE and kNN notably declines when missingness is ≥ 30% (Penone et al. 2014; Taugourdeau et al. 2014). Therefore, our results for 'OrdKrig', compared to those for 'mice' and 'kNN', show that  spatial structure, rather than trait covariation, may provide more accurate trait imputations when gaps are frequent (Fig. 2, Fig. S7, S8).**

- However, it should be taken into account that the focus of the paragraph is rather to look at generalities, not at differences between traits. We will discuss this issue in more detail in the new version when looking at the role of auxiliary variables in improving imputations (and how

these improvements depend specifically on the trait and type of auxiliary variable). For an example, see our reply to comment *R2#24*.

*R2#17. L239, L246, L250: Instead of Supplementary S5, please indicate Fig. S number.*
*Please do so for the rest of the paper as well.*
- Done.

*R2#18. L246 Please remove "however".*
- Done.

*R2#19. L279: "stand structure" was mentioned before (L25, L90, L146, etc.) as "forest structure".*
*Please choose one and be consistent.*

- We have replaced 'stand structure' by 'forest structure' (three appearances throughout the text)

*R2#20. Chapter 3.2: Why the comparison of adding different levels of auxiliary information was only used for MICE and not for kNN or RegKrig?*

- We actually did this exercise mainly for MICE, but also to some extent for kNN, because we ran kNN using species (kNN_s) and the best predictor set (kNN_ctsp). We used these two methods because they were easier to implement in our simulation framework and they are both algorithms that are designed to deal with multivariate missing data (as opposed to regression kriging). Moreover, the study already includes many different aspects (imputation methods, auxiliary information used in the imputations, missingness, multiple traits) and we think that our approach already shows, for a subset of imputation methods, the role of different auxiliary variables in improving the imputations, for different traits.

*R2#21. In addition, please discuss why mice_ctsp was chosen while according to Fig. 4 mice_sc would be sufficient for Nmass, LMA and WD and mice_st for Hmax and only BL:AS needs mice_cstp (extra p).*

- We agree with the reviewer in that, for some traits, imputation performance does not improve in mice_ctsp compared to some mice applications with less auxiliary variables. However, we had to choose a number of auxiliary variables for which performance was maximised for all traits. This is the reason why we chose mice_ctsp. We have now specified this in the text (section 3.2):
- **Our results collectively suggest that, apart from species identity, different types of ecological information, particularly forest structure and topography, may improve statistical imputation schemes. In contrast, the role of climate, lithology and sampling month in improving imputations was comparatively minor. However, we selected 'mice_ctsp' as the method that performed best for all traits, because adding climate did not reduce imputation performance and not including 'topography' would worsen BL:AS imputations.**
- Note also that using different levels of auxiliary information for different traits would complicate substantially the comparison of different methods (the main objective of the paper) and the interpretation of the results.

*R2#22. L288: Please replace A with S in "Fig. A7 and A12".*

- Done

*R2#23. L288-289: what about adding only species and forest structure to RegKrig? Climate and topography should be already accounted for using spatial information in RegKrig.*

- We could not include species identity for regKrig because, for less common species, there were not enough data to perform the regKrig imputations at high missingness rates. This is also one of the reasons to use methods that are designed to handle multivariate missing datasets, such as MICE or kNN.

*R2#24. L292-296: Please discuss and reason the differences of the results for the different traits.*

- Please note that our main objective is to compare different methods and different levels of environmental information for a diverse set of widely used traits. Discussing in detail differences in performance between traits would go much beyond the scope of the paper and would tend to blur our main, more general conclusions, which are already complex as a result of the comprehensive nature of the paper, in which we test many different aspects related to imputation performance. Nevertheless, we will try to discuss specific differences among traits when particularly relevant in the context of our study. For example, we have discussed the patterns we observe here with those by Vilà-Cabrera et al 2015, here in section 3.2:
- **Introducing auxiliary variables as predictors improved MICE performance substantially but these improvements were dependent on the specific predictor set and trait (Fig. 4, Fig. S10). Species identity increased KGE for all traits (Fig. 4) and it was the major predictor for Nmass, LMA and WD, as all MICE applications with species identity performed significantly better than those not including it (Fig. 4; P<0.05). Forest structure notably improved imputations for Hmax and for BL:AS at missingness ≥ 50% (P <0.05). Climate only produced significant increases in KGE (i.e., compare 'mice' with 'mice_c' in Fig. 4) for Hmax and WD (P<0.05). These results are in line with the distinct role of phylogeny and environmental variables as drivers of trait variability recently observed for the same tree species in the IEFC (Laforest-Lapointe et al. 2014; Vilà-Cabrera et al. 2015). After controlling for family (Pinaceae and Fagaceae), environmental variables only explained a substantial fraction of the variability for Hmax, they explained very little variability for LMA and WD and played no role in explaining Nmass (Vilà-Cabrera et al. 2015).**

- And in section 3.3:
- **In terms of KGE, 'mice_ctsp' was the best performing method at 50% missingness for all traits, together with 'Spmean' for Nmass,and LMA and with 'RegKrig' for Hmax (P>0.05 for comparisons between 'mice_ctsp', 'Spmean' and 'RegKrig for these traits). However, at 80% missingness, 'mice_ctsp' only ranked first for BL:AS whereas 'Spmean' showed the highest KGE for Nmass, LMA and WD and 'RegKrig' performed best for Hmax (Fig. 6, Fig. S12). These results are consistent with the prominent role of taxonomic identity in explaining variability in foliar traits and WD and with the higher predictive ability of environmental and spatial information in explaining Hmax (Vilà-Cabrera et al. 2015).**

*R2#25. L305: "(data not shown here)". But why not adding a figure similar to Fig. S8, for the comparison of ∆cormat and dataset NRMSE between Spmean, mice_ctsp, kNN_ctsp and RegKrig in the supplementary as well?*

- We had in fact this figure in a preliminary draft and we decided to leave it out. We will reduce supplementary materials in the revised version (from 32 pages to ~ 19 pages, as also suggested by reviewer #1) and will only provide the result of the corresponding statistical test,

and we will not add a new figure. Here's the text in its modified version although supplementary materials have not been reduced yet:

- **Kernel density plots and Kolmogorov-Smirnov tests (Fig. S18, S19) showed that MICE produced imputations (especially 'mice_ctsp') most consistent with observed distributions at all missingness levels (Fig. S18, S19). 'Spmean' and 'OrdKrig' imputations modified trait distributions substantially, while 'kNN_ctsp' and 'RegKrig' showed an intermediate performance, but generally far from that of 'mice_ctsp' (Fig. S22, Table S1, S2). 'Spmean' and kriging imputations also yielded larger Δcormat values compared to the rest of the methods (P<0.05).**

*R2#26. L307-315: I would suggest to make this part chapter 3.4 (related to "Imputing traits for the main forest species in Catalonia 2.6)*

*\*\*IMPLICATIONS\*\**
*R2#27. This part is well written. I would suggest to add also suggestion for improvement of data collections, e.g. trait collection should be better be accompanied by auxiliary information on coordinates, forest structure, etc.*

- Thanks for this suggestion. We have included this idea in the 'Implications' section, where we also mention that databases of vegetation structure could help improve trait imputation:
- **For kNN, MICE and kriging imputations we have highlighted the key role of auxiliary variables as necessary covariates to yield reliable imputations in spatially explicit settings. This result calls for the inclusion of site-specific environmental variables associated with trait data in trait databases. The importance of covariates differed across traits, but, in addition to the expected influence of species, climate and topography in predicting trait values, we also showed a prominent role of forest structure for some traits. The ongoing development of global databases of vegetation structure (e.g. Dengler et al. 2014) will likely enable the incorporation of stand variables in trait imputation approaches using spatial and environmental information (Butler et al. 2017).**

    **New reference:**
- **Butler, E. E., Datta, A., Flores-Moreno, H., Chen, M., Wythers, K. R., Fazayeli, F., Banerjee, A., Atkin, O. K., Kattge, J., Amiaud, B., Blonder, B., Boenisch, G., Bond-Lamberty, B., Brown, K. A., Byun, C., Campetella, G., Cerabolini, B. E. L., Cornelissen, J. H. C., Craine, J. M., Craven, D., Vries, F. T. de, Díaz, S., Domingues, T. F., Forey, E., González-Melo, A., Gross, N., Han, W., Hattingh, W. N., Hickler, T., Jansen, S., Kramer, K., Kraft, N. J. B., Kurokawa, H., Laughlin, D. C., Meir, P., Minden, V., Niinemets, Ü., Onoda, Y., Peñuelas, J., Read, Q., Sack, L., Schamp, B., Soudzilovskaia, N. A., Spasojevic, M. J., Sosinski, E., Thornton, P. E., Valladares, F., Bodegom, P. M. van, Williams, M., Wirth, C. and Reich, P. B.: Mapping local and global variability in plant trait distributions, PNAS, 114(51), E10937–E10946, doi:10.1073/pnas.1708984114, 2017.**

*R2#28. L322: "Here we deal here" Please remove the extra "here".*
- Done

*R2#29. L357: Please change "practices" to "practiced".*
- We think this is correct here, as we refer to the noun 'practices'.

*\*\*SUPPLEMENTARY\*\**
*R2#30. Please refer to the supplementary with their Fig. numbers and not chapter numbers.*
- Done.

*R2#31. Please make sure that all supplementary figures are mentioned in the main text. Currently this is not the case.*
- Thanks for spotting this. We will check this once we reduce the figures in the supplementary materials.

R2#32. *Fig S7: trait mean (Mean) is missing in the plot.*
- This is because KGE is undefined for mean imputations, as there is no variability in the imputed variables.

*R2#32. Fig S2: Please change "palant" to "plant".*
- Done

*R2#33. Fig S6: Please define "cor.matrix abs.error" in the figure and "correlation matrix error"in the caption. Do you mean ∆cormat? Please be consistent.*
- Yes, it should be ∆cormat, thanks for noticing this. We will change it in the revised version.

---

## Author Response (AR1)

Dear Dr. Thonicke,

Thanks for carefully looking at our replies to the referees and for your very positive assessment of our paper. Please find below the reply to your comments and suggestions.

*- Original comments by the editor will be shown in italics*
- Author's reply will be shown in normal font
**- Actual changes in the manuscript will be shown in bold.**

We also provide a point-by-point list of changes, when applicable, with line numbers corresponding to the revised version of the manuscript (no track changes), and mapped to the original comments by the reviewers (see our published author comments).

Finally, any (minor) issue we have detected during the revision has been corrected and listed at the end of this document (see 'Other changes).

Best regards,

Rafael Poyatos, on behalf of the coauthors

Reply to Editor's comments

*I have minor suggestions to make for the fully revised manuscript:*
*R1#3:*
*1. Please merge the supplementary figures and merge sections S3 and 4 as you suggested.*
*2. And yes, please move Table S2 to the main text to make it Table 1 and modify the text accordingly.*
*3. Please insert your suggestion for Figure 2 to illustrate the multiple comparisons*

> We have made all the suggested changes, but for number 3 in the list above, please note that we have only added the letters for the multiple comparisons in Figs. 2, 3, 6 and 7 (not in Figs. 4 or 5) because of lack of space. The P-values for the relevant comparisons concerning these figures have been added to the main text. Please note that letters denote results of multiple comparisons, in alphabetical order from highest to lowest performance. This clarification has been added to each figure caption showing these comparisons.

*R1#9: Please check how you can include your response to this reviewer point in the manuscript. Also to emphasize what you regard as being unique.*

> We have now included the reasoning in our reply to R#19 in the main text:

> **Assessments of imputation methods in the ecological literature have not tested the impact of the choice of univariate imputation models within MICE (Penone et al. 2014, Taugourdeau et al. 2014). Here we showed that predictive mean matching (PMM), the default algorithm in the mice package, performed comparably well compared to alternative methods (Supplement S3, Fig. S3, S4). Therefore, we used MICE with PMM as the univariate imputation model, also because it is robust to non-normality and preserves non-linear relationships between variables (Morris et al., 2014).**

*R1#17: Please check if you can include a sentence on the number of iterations in the method section in addition to the added sentence in response to R1#18*

> We have added a sentence justifying the number of iterations:

> **We set t = 2 to ensure convergence and to minimise the effects of imputation order (van Buuren 2012).**

*R2#3 Please check the entire manuscript text that you consistently use auxiliary information or variable as suggested by the reviewer*

We now use the term 'auxiliary variables' throughout the paper when referring generically to variables that are used as predictors in imputation processes. However, please note that to refer to the specific variables we have tested in this paper, we now use the term' 'environmental information', following the comment R1#11 by the first reviewer. See, for example, the change in the title.

*Both reviewers asked why species identity was not part of your analysis. Please check if you can include this point in implications or elsewhere in the discussion.*

We have changed the opening paragraph of the 'Implications' section to include our reasoning on the lack of species-specific results:

**The problem of missing data is ubiquitous in plant trait datasets of regional to global scope. Recently, ecologists have made substantial progress in (i) the assessment of the best imputation methods in trait-based applications, (ii) how these methods perform with increasing missingness, (iii) which ecological covariates aid to improve imputations and (iv) how different imputation methods impact the results of trait-based analyses (Pakeman, 2014, Taugourdeau et al. 2014, Penone et al. 2014, Schrodt et al. 2015). Most effort thus far, however, has been directed at imputing species-level trait means and all the abovementioned questions have rarely been assessed on the same dataset. Here we deal with all the previous issues simultaneously and also deal withthe spatial component of trait variability, where the intra-specific component cannot be neglected. We did not focus on differences in imputation errors across species because this issue is, to a large extent, related to the degree of trait variability explained by biotic and abiotic predictors across different taxa, which was recently reported by Vilà-Cabrera et al. (2015).**

*Apart from those mentioned in my list above, I ask you to please include all suggested modifications to the manuscript as you described them in the response letters.*

Please find at the end of this document a list with the most substantial changes, with a reference to the corresponding referee comment and the lines of the current manuscript where changes have been made.

*I think your manuscript will greatly profit from the detailed reviews thanks to the time both reviewers invested.*

We would like to thank the editor and the reviewers for the constructive comments on our manuscript.

**List of changes**

Reviewer #1

R1#2. See L. 140-145.
R1#3.
- In the supplement, the original Figs. S2, S4, S7, S10, S12, S13-S17 have been removed. Original sections S3 and S4 in the Supplementary Materials have been merged and the original section S7 (quantification of the effect of missingness levels) has been integrated in current section S6.
- Original Table S2 now is Table 1.
- Changes in section 3.3, L. 326 - 330.

- See the addition of letters depicting multiple comparisons in Fig. 2, 3, 6, and 7 and the reporting of P-values for meaningful comparisons throughout the text (L. 260, 261, 273, 271, 281, 282, 290, 298, 300, 304, 317, 318, 322, 331, 337, 340, 342, 343) . See also changes in the 'Methods' section (L. 243-248).
R1#5. See changes in L. 406-419.
R1#7. See changes in L. 380-385.
R1#8. See changes in L. 56-62.
R1#9. See changes in L. 169-174.
R1#10. L. 68-70.
R1#11. See changes in L. 74-78.
R1#13. See changes in L. 376-379.
R1#14. See changes in L. 346-351.
-See changes in L. 425-427.
R1#15. See changes in L. 406-408.
R1#18. See L. 158-159.
R1#19. Original sections S3 and S4 in the Supplementary Materials have been merged. Also, the current explanation of MICE in the main text includes more details (e.g. number of iterations, justification of PMM). See L. 164-178.
R1#20. See changes in L.193-196.
R1#22. All figures showing KGE now have a caption clarifying that higher KGE means better performance (Figs. 4, 6).
R1#25. This comment refers to the reduction of the Supplementary Materials. See R1#3 above.
R1#26. See changes in L. 72-74 of the Supplementary Material S3.
R1#27. Original sections S3 and S4 in the Supplementary Materials have been merged.
R1#28
R1#29. Full description of the traits have been added to the captions in Figs. 2-8 and Figs. S3, S6, S8-S13).
R1#30.
R1#31
R1#32. See changes in L. 257-266.
R1#33. In response to the reviewer's comments, see the changes in the following sections of the manuscript:
- L. 257-259.
- L. 269-272
R1#36. See new section added (3.4, L. 353).
R1#37. We now provide several statistical tests in Fig. 2, 3, 6, 7, Tables 1 and S1, and throughout the text (L. 260, 261, 273, 271, 281, 282, 290, 298, 300, 304, 317, 318,  322, 331, 337, 340, 342, 343).
R1#38. L. 181-182.
R1#40. See our changes in R1#37.
R1#41. We now refer to figures in the supplement by the number (L. 106, 159,172, 176, 259, 266, 269, 271, 272, 274, 275, 276, 299, 328, 333, 334, 336, 347, 349, 358, 359, 363).
R1#42. See our changes in R1#29.

Reviewer 2

R2#2. Full description of the traits have been added to the captions in Figs. 2-8 and Figs. S3, S6, S8-S13).
R2#3. We now use the term 'auxiliary variables' throughout the paper when referring generically to variables that are used as predictors in imputation processes (see L. 77, 79, 150,  159, 185, 186, 201, 252, 278, 302, 303, 314, 318, 361, 390 ). However, please note that to refer to the specific variables we have tested in this paper, we now use the term 'environmental information'.
R2#5. See L. 26-29.
R2#6. See L. 46-47.
R2#8. See L. 96-97.
R2#9. See L. 140-142.
R2#10. See L. 126-128.
R2#11. See L. 189.
R2#13. See changes in L. 213-214, L. 230-232, L. 238-241, L. 225-227.
R2#14. See L. 252.
R2#16. See L. 257-266.
R2#17. We now refer to figures in the supplement by the number (L. 106, 159,172, 176, 259, 266, 269, 271, 272, 274, 275, 276, 299, 328, 333, 334, 336, 347, 349, 358, 359, 363).
R2#18. See L. 269.

R2#19. See changes in L. 27, 96, 111, 150, 187, 191, 193, 196, 281, 296, 303, 307, 317, 395, 601, 628, 635, 643, 649, 655).
R2#21. See changes in L. 306-310.
R2#22. The change proposed in the author comment does not apply anymore, as this text has been deleted.
R2#24. See changes in L. 278-287, L. 321-326.
R2#25. See changes in L. 333-337.
R2#26. See new section 3.4 (L. 353)
R2#27. See changes in L. 391-397.
R2#28. See L. 371.
R2#30. See changes in R2#17 above.
R2#31. All supplementary figures are now cited in the main text.
R2#33. This mistake has been corrected in current figures S2 and s3.

**Other changes**
- The figure that is now Fig. S3 was not the correct one in the previous version of the manuscript, and we have now fixed this.

[revised manuscript text omitted]